# Identification of a lipid scrambling domain in ANO6/TMEM16F

Kuai Yu[1]*[†], Jarred M Whitlock[1][†], Kyleen Lee[1], Eric A Ortlund[1,2], Yuan Yuan Cui[1], H Criss Hartzell[1]*

[1]Department of Cell Biology, Emory University School of Medicine, Atlanta, United States; [2]Department of Biochemistry, Emory University School of Medicine, Atlanta, United States

**Abstract** Phospholipid scrambling (PLS) is a ubiquitous cellular mechanism involving the regulated bidirectional transport of phospholipids down their concentration gradient between membrane leaflets. ANO6/TMEM16F has been shown to be essential for $Ca^{2+}$-dependent PLS, but controversy surrounds whether ANO6 is a phospholipid scramblase or an ion channel like other ANO/TMEM16 family members. Combining patch clamp recording with measurement of PLS, we show that ANO6 elicits robust $Ca^{2+}$-dependent PLS coinciding with ionic currents that are explained by ionic leak during phospholipid translocation. By analyzing ANO1-ANO6 chimeric proteins, we identify a domain in ANO6 necessary for PLS and sufficient to confer this function on ANO1, which normally does not scramble. Homology modeling shows that the scramblase domain forms an unusual hydrophilic cleft that faces the lipid bilayer and may function to facilitate translocation of phospholipid between membrane leaflets. These findings provide a mechanistic framework for understanding PLS and how ANO6 functions in this process.

*For correspondence: kyu3@
emory.edu (KY); criss.hartzell@
emory.edu (HCH)

[†]These authors contributed
equally to this work

**Competing interests:** The
authors declare that no
competing interests exist.

**Reviewing editor**: Randy
Schekman, Howard Hughes
Medical Institute, University of
California, Berkeley, United
States

## Introduction

The transbilayer asymmetric distribution of phospholipids in cell membranes is evolutionarily conserved and essential to cellular physiology. In eukaryotic plasma membranes, the outer leaflet is enriched in phosphatidylcholine and sphingomyelin and the inner cytoplasmic-facing leaflet is rich in phosphatidylserine (PtdSer) and phosphatidylethanolamine (PtdEtn) (*Bretscher, 1972*; *Fadeel and Xue, 2009*; *Lhermusier et al., 2011*; *van Meer, 2011*). This asymmetry is established by ATP-dependent lipid flippases, most notably members of the P4 ATPase family (*Panatala et al., 2015*) and ABC transporters (*Borst et al., 2000*), that actively transport phospholipids to one leaflet of the membrane and are important in membrane biogenesis. Phospholipid asymmetry plays critical roles in membrane function in two ways. (1) Charges on the phospholipid head groups bind and regulate protein function. The best known examples may be the effects of phosphatidylinositol bisphosphate on ion channel proteins (*Suh and Hille, 2008*), but PtdSer and PtdEtn play equally important roles (*Fairn et al., 2011*; *Jeong and Conboy, 2011*; *Kay and Grinstein, 2013*; *Hosseini et al., 2014*). (2) The different molecular shapes (e.g., cylindrical or conical) of various lipids determine membrane curvature which is key to membrane trafficking and fusion (*Graham and Kozlov, 2010*; *Bigay and Antonny, 2012*; *Xu et al., 2013*; *Suetsugu et al., 2014*).

In opposition to ATP-dependent flippases, ATP-independent phospholipid scramblases (PLSases) facilitate the equilibration of phospholipid distribution between the two membrane leaflets. PLSases play essential roles in the synthesis of glycoconjugates, such as N-glycosylated proteins and GPI-anchored proteins in the endoplasmic reticulum (*Pomorski and Menon, 2006*). Moreover, at the plasma membrane dissipation of phospholipid asymmetry by phospholipid scrambling (PLS) is a common cell signaling mechanism (*Fadeel and Xue, 2009*; *Bevers and Williamson, 2010*). For example, exposure of

**eLife digest** Cell membranes are made of two layers of molecules called phospholipids. The types of phospholipid molecules in the outer layer are often different from those in the inner layer. This asymmetry is an important feature of most membranes, but cells also have proteins called 'scramblases' that can move (or scramble) the phospholipids between the two layers. This scrambling process often marks a cell for destruction but also plays a key role in many activities throughout the body including cell–cell fusion, blood clotting, autoimmune diseases and inflammation.

Previous research revealed that a membrane protein called ANO6 is needed for some kinds of phospholipid scrambling. Other proteins that are most closely related to ANO6 are not phospholipid scramblases; instead they are channel proteins that allow ions to pass across cell membranes. ANO6 can also allow ions to flow across membranes, which raised the question: is ANO6 actually a scramblase itself, or does it control other proteins with scramblase activity?

Yu, Whitlock et al. addressed this question by engineering human cells grown in the laboratory to produce the ANO6 protein, and found that these cells had high levels of phospholipid scrambling. Next, the scrambling of phospholipids in these cells was measured while the flow of ions through ANO6 was also recorded. These experiments revealed that these two processes happened almost simultaneously. Yu, Whitlock et al. suggest that this could mean that ANO6 allows ions to leak through when it shuttles phospholipids between layers of cell membranes.

ANO1 is an ion channel that is related to ANO6 but it does not have scramblase activity. By designing and testing hybrid proteins that combined parts of ANO6 and ANO1, Yu, Whitlock et al. identified the part of ANO6 that is responsible for its scramblase activity. Furthermore, computer models of this 'scrambling domain' suggest that it forms an unusual groove that faces into the cell membrane, and that could help phospholipids to shuttle between the inner and outer layers of the membrane. Alternatively, this groove could interact with other proteins to regulate phospholipid scrambling; and if so, further work will be needed to identify these unknown proteins.

Finally, swapping a relatively small number of features between ANO6 and ANO1 could confer scrambling activity on ANO1. This suggests that ANO1 may itself have a special relationship to membrane phospholipids. Uncovering the nature of this relationship, if it exists, as well as understanding how ANO6 scrambles phospholipids will challenge structural biologists to generate high-resolution images of these proteins in complex with phospholipids.

PtdSer on the external leaflet of the plasma membrane marks apoptotic cells for phagocytosis by macrophages and plays a key role in blood clotting (*Fadok et al., 1992*; *Verhoven et al., 1995*; *Emoto et al., 1997*; *Fadok et al., 2001*; *Suzuki et al., 2010*; *Lhermusier et al., 2011*; *Kay and Grinstein, 2013*). PtdSer exposure that occurs when platelets sense tissue damage serves as a catalytic surface for assembly of plasma-borne coagulation factors and a $>10^6$-fold increase in the rate of thrombin formation (*Zwaal et al., 1998*; *Sahu et al., 2007*; *Kay et al., 2012*). PtdSer exposure also plays important roles in developmental processes that involve fusion of mononucleated progenitor cells to form multinucleated cells such as skeletal muscle (*Jeong and Conboy, 2011*; *Hochreiter-Hufford et al., 2013*), osteoclasts (*Pajcini et al., 2008*; *Helming and Gordon, 2009*; *Harre et al., 2012*; *Shin et al., 2014*; *Verma et al., 2014*) and placental syncytiotrophoblasts (*Huppertz et al., 2006*; *Riddell et al., 2013*).

In the simplest conceptualization, PLS is mediated by PLSases that are thought to provide, in a manner analogous to ion channels, an aqueous pathway for the hydrophilic phospholipid head groups to flip between the inner and outer leaflets (*Pomorski and Menon, 2006*; *Sanyal and Menon, 2009*). However, the molecular mechanisms of PLS have remained elusive partly because PLS can be catalyzed by a variety of unrelated proteins. PLS is stimulated by two pathways, a rapid one triggered by increases in intracellular [Ca$^{2+}$] and a slow caspase-dependent pathway associated with apoptosis (*Schoenwaelder et al., 2009*). At least four different families of proteins have been implicated in PLS (*Sahu et al., 2007*; *Bevers and Williamson, 2010*; *Lhermusier et al., 2011*). The first putative PLSases (PLSCR1–PLSCR4) were identified by their ability to stimulate Ca$^{2+}$-dependent PtdSer scrambling when incorporated into liposomes (*Basse et al., 1996*; *Comfurius et al., 1996*), but the role of PLSCRs in PLS is controversial because these proteins are small calmodulin-like molecules whose

disruption in mice, flies, and worms has little effect on PLS (*Acharya et al., 2006*; *Fadeel and Xue, 2009*; *Bevers and Williamson, 2010*; *Ory et al., 2013*). More recently, the Xk-family protein Xkr8 has been shown to be necessary for caspase-dependent PtdSer scrambling that can be rescued by multiple Xkr8 paralogs (*Suzuki et al., 2013a*, *2014*). Unexpectedly, several G-protein coupled receptors including rhodopsin and the β-adrenergic receptor have also been shown to elicit PLS (*Goren et al., 2014*). Although lipid-translocating ABC transporters and phospholipid synthesis may also play roles in PLS, these processes cannot fully explain PLS (*Fadeel and Xue, 2009*; *Goren et al., 2014*).

Recently, it has been suggested that some members of the 10-gene Anoctamin (TMEM16) family are PLSases. Evidence supporting a role for ANO6 in PLS includes the findings that (a) mutations in ANO6 produce Scott Syndrome, a blood clotting disorder where platelets fail to expose PtdSer in response to cytosolic $Ca^{2+}$ increases (*Suzuki et al., 2010*; *Yang et al., 2012*; *Kmit et al., 2013*), (b) knockout of ANO6 in mice abolishes the ability of cells to expose PtdSer and to scramble other lipid species in response to elevated cytosolic $Ca^{2+}$ while PLS stimulated by the apoptotic Fas receptor is unaffected (*Suzuki et al., 2010*), and (c) over-expression of ANO6 in knockout cells rescues PLS (*Suzuki et al., 2013b*). However, the suggestion that ANO6 is a PLSase is confounded by the fact that other members of the anoctamin family, ANO1 and ANO2, encode the pore-forming subunits of $Ca^{2+}$-activated $Cl^-$ channels (CaCCs) (*Caputo et al., 2008*; *Schroeder et al., 2008*; *Yang et al., 2008*; *Hartzell et al., 2009*; *Duran and Hartzell, 2011*; *Pedemonte and Galietta, 2014*; *Ruppersburg and Hartzell, 2014*). It was initially presumed that all ANOs are $Cl^-$ channels because of the close sequence similarity between ANO family members (ANO1 is 38–45% identical to ANOs −3 to −7, >90% coverage), however, the requirement of ANO6 for $Ca^{2+}$-dependent PLS and the ability of ANOs 3, 4, 6, 7, and 9 to rescue this activity suggests functional divergence within the ANO family (*Suzuki et al., 2010*, *2013b*).

Questions remain whether ANO6 is a PLSase itself and/or is an ion channel that regulates PLSase activity (*Kunzelmann et al., 2014*; *Pedemonte and Galietta, 2014*; *Picollo et al., 2015*). ANO6 has been reported to be a non-selective cation channel (*Yang et al., 2011*; *Adomaviciene et al., 2013*), a swelling-activated $Cl^-$ channel (*Almaca et al., 2009*), an outwardly-rectifying $Cl^-$ channel (*Martins et al., 2011*), a CaCC (*Szteyn et al., 2012*; *Shimizu et al., 2013*; *Juul et al., 2014*), and a CaCC of delayed activation (*Grubb et al., 2013*). Furthermore, in contrast to *Suzuki et al. (2010)*, *Yang et al. (2012)* conclude that ANO6 is a regulator of an endogenous PLSase because they find that expression of ANO6 in HEK cells does not cause PtdSer exposure. However, the suggestion that some ANOs are PLSases and are not simply regulators of endogenous PLSases is supported by recent reports that two fungal ANO homologs purified and incorporated into liposomes mediate $Ca^{2+}$-stimulated PLS (*Malvezzi et al., 2013*; *Brunner et al., 2014*).

Here we address three questions. (1) Does expression of ANO6 stimulate $Ca^{2+}$-dependent PLS in HEK cells? We find that ANO6 expression in HEK cells induces robust $Ca^{2+}$-activated scramblase activity. (2) Is the ion channel activity of ANO6 related to PLS? We find that ANO6 currents are non-selective among ions and are activated simultaneously with PLS. This suggests that ionic currents are a consequence of phospholipid translocation. Chimeric constructs that exhibit PLS also exhibit simultaneous non-selective currents. Furthermore, we find that drugs that block ANO1 currents do not block ANO6 currents or PLS. This is consistent with the idea that the ion conduction pathway associated with ANO6 is unlike the ANO1 pore. (3) What are the domains of ANO6 that are required for PLS? We find that mutating several amino acids in ANO6 between TMD4 and TMD5 eliminates both PLS and ion channel activity. Furthermore, replacing as few as 15 amino acids in ANO1 in the TMD4-TMD5 region with ANO6 sequence confers robust PLS activity on ANO1, which normally does not elicit PLS activity.

## Results

### ANO6 expression induces robust PLS in HEK cells

We first asked whether ANO6 when expressed heterologously induced PLS in HEK293 cells. $Ca^{2+}$-dependent PtdSer exposure on the outer leaflet of the plasma membrane was measured by confocal imaging of two PtdSer probes, either LactoglobulinC2 fused to Clover fluorescent protein ('LactC2') or Annexin-V conjugated to AlexaFluor-568 ('Annexin-V') (*Shi et al., 2006*; *Hou et al., 2011*; *Kay and Grinstein, 2011*). Our initial experiments employed LactC2 with a clonal cell line stably transfected with mANO6-FLAG$_{3X}$. 100% of the cells express ANO6-FLAG$_{3X}$ as shown by staining with

anti-FLAG antibody (*Figure 1A*). Intracellular $Ca^{2+}$ was elevated by incubation of the cells in 10 μM A23187 in nominally zero $Ca^{2+}$ solution followed by washout of A23187 and addition of 5 mM $Ca^{2+}$ to initiate $Ca^{2+}$-dependent PLS. 12 min after adding $Ca^{2+}$, ~92% of the cells (N = 404) showed LactC2 binding to the surface (*Figure 1B,G*). In contrast, little or no LactC2 binding was seen in the parental cell line (0.2% positive cells, N = 724) or in ANO1-FLAG$_{3X}$-expressing cells (8% positive cells, N = 316) (*Figure 1C,D,G*). The difference in PLS observed with ANO1 and ANO6 expressing cells was not explained by differences in expression as shown by western blot (*Figure 1F*). These data show clearly that ANO6 expression facilitates PLS. However, the finding that a small fraction (8%) of ANO6-FLAG$_{3X}$

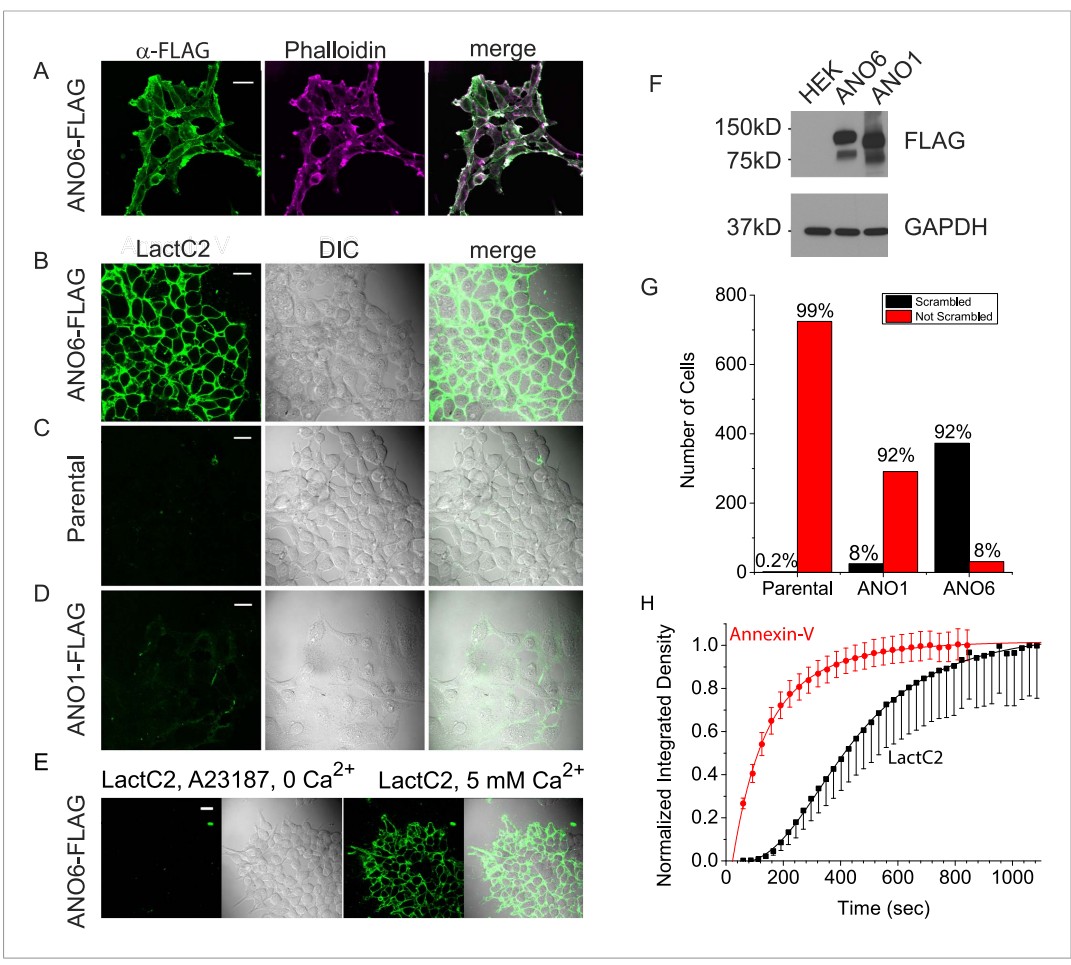

**Figure 1**. Expression of ANO6 in HEK cells stimulates $Ca^{2+}$-dependent phospholipid scrambling (PLS). (**A**) Images of HEK cells stably transfected with ANO6-FLAG$_{3X}$ fixed and stained with anti-FLAG (green) and phalloidin (magenta). (**B–D**) Images of live cells after exposure to 10 μM A23187 in zero-$Ca^{2+}$ solution for 5 min followed by solution containing 5 mM $Ca^{2+}$ and 3 μg/ml LactC2-Clover for 12 min as described in 'Materials and methods'. Green channel: LactC2-Clover. DIC channel: differential interference contrast. (**B**) HEK cells stably expressing ANO6-FLAG$_{3X}$. (**C**) Parental HEK cells not expressing ANO6. (**D**) HEK cells stably transfected with ANO1-FLAG$_{3X}$. (**E**) A23187 in zero $Ca^{2+}$ does not stimulate LactC2 binding. The first two panels show LactC2-Clover binding in cells incubated in A23187 in zero $Ca^{2+}$ containing LactC2 for 15 min. The second two panels show the same cells 10 min after adding 5 mM $Ca^{2+}$. (**F**) Level of expression of ANO1 and ANO6 in stably expressing HEK cells. Extracts of cells in **B–D** were immunoblotted with anti-FLAG and anti-GAPDH to quantify protein expression. (**G**) Numbers of cells binding Annexin-V ('scrambled') or not binding Annexin-V ('not scrambled') for parental HEK cells, ANO6-FLAG$_{3X}$, and ANO1-FLAG$_{3X}$ expressing cells. (**H**) Time course of Annexin-V and LactC2 binding to HEK cells expressing ANO6-FLAG$_{3X}$. Images of the same field of 30–100 cells were acquired at ~20 s intervals. Annexin-V: mean ± SEM of seven independent experiments, LactC2: mean ± SEM of five independent experiments. Means at the end of the recordings were normalized to 1. Scale bars = 20 μm.

expressing cells do not exhibit PLS raises the possibility that ANO6 may not be sufficient for PLS and may require additional components.

An advantage of LactC2 is that it does not require $Ca^{2+}$ to bind PtdSer (*Shi et al., 2006*; *Kay and Grinstein, 2011*), so we were able to use it to test whether ANO6-elicited PtdSer exposure requires $Ca^{2+}$. ANO6-FLAG$_{3X}$ cells exposed to A23187 without addition of $Ca^{2+}$ exhibited no detectable LactC2 binding over 15 min, whereas subsequent addition of $Ca^{2+}$ stimulated robust LactC2 binding (*Figure 1E*).

Next, we compared LactC2 and Annexin-V as probes for PLS. The percentage of cells stained with Annexin-V and LactC2 were the same, however, the kinetics of binding of the two probes were markedly different. After elevating cytosolic $Ca^{2+}$, Annexin-V fluorescence increased mono-exponentially with a mean $\tau$ = 143 s (7 separate experiments, $\chi^2$ of fit = 0.03) and approached a plateau within less than 10 min (*Figure 1H*). In contrast, the time course of LactC2 binding was significantly slower (5 separate experiments, $\tau$ = 433 s, $\chi^2$ of fit = 0.02). LactC2 binding exhibited a lag period of 1–2 min before binding was detectable. One potential explanation of the difference in Annexin-V and LactC2 time courses may be related to the fact that LactC2 prefers binding to oxidized PtdSer (*Tyurin et al., 2008*). Because Annexin-V binds with more rapid kinetics, the rest of the experiments shown here were performed using Annexin-V.

Because all of the cells in the clonal cell line express ANO6-FLAG$_{3X}$ at about the same level, we turned to a polyclonal cell line stably expressing ANO6-EGFP to evaluate the relationship of ANO6 expression to the rate and extent of Annexin-V binding (*Figure 2*). In the polyclonal line, ANO6-EGFP expression level was variable and 85% of ANO6-EGFP cells exhibited PLS (N = 180). This percentage may be lower than the ANO6-FLAG$_{3X}$ clonal line because some ANO6-EGFP positive cells may have lower expression than the ANO6-FLAG$_{3X}$ cells. There was a direct relationship between the level of ANO6-EGFP expression and the level of Annexin-V binding 10 min after elevating cytosolic $Ca^{2+}$ (Pearson correlation coefficient = 0.84) (*Figure 2C*), however there was considerable variation around the fitted relationship, with some highly-expressing ANO6 cells showing little Annexin-V binding and low-expressing cells showing high levels of binding. The rate of Annexin-V binding was similar among cells ($\tau$ in this experiment ranged from 152–229 s), but the plateau level of Annexin-V binding attained after 12 min varied markedly among cells. Neither the rate nor the plateau level of binding correlated with the level of ANO6-EGFP fluorescence. This lack of correlation may reflect an inability to distinguish between ANO6-EGFP located on vs adjacent to the plasma membrane or may reflect heterogeneity in the expression of other components required for PLS.

## ANO6 current activates in parallel with PLS

The next question that we sought to answer was whether the ionic current that has been associated with ANO6 (*Almaca et al., 2009*; *Martins et al., 2011*; *Yang et al., 2011*; *Szteyn et al., 2012*; *Grubb et al., 2013*; *Harper and Poole, 2013*; *Shimizu et al., 2013*; *Juul et al., 2014*) is linked to PLS or is an independent function of the protein. We patch-clamped HEK cells transiently expressing ANO6-EGFP and recorded ANO6 currents while simultaneously imaging Annexin-V binding. A typical experiment is shown in *Figure 3A* with average results in *Figure 3B,C*. After establishing whole-cell recording, the time course of activation of membrane currents in ANO6-expressing cells was very slow even when the patch pipet solution contained high (200 μM) $Ca^{2+}$. The currents typically began to increase 8 min after initiating whole cell recording, which is similar to that reported by others (*Grubb et al., 2013*). Annexin-V binding usually became detectable 2–3 min later (*Figure 3B,C*). The time courses of ANO6 current activation and Annexin-V binding were fit to the equation $y = A2 + (A1 - A2)/1 + \exp[(t - t_0)/\tau]$. Although the time constants of the increases in Annexin-V binding ($\tau$ = 2.24 ± 0.15 min) and ANO6 current ($\tau$ = 2.31 ± 0.34 min) were the same, the time ($t_0$) at which Annexin-V fluorescence reached half of its maximal value [(A1 + A2)/2] was delayed 3 min relative to the current. In contrast to ANO6, ANO1 currents activated quickly after initiating whole-cell recording and then ran down with time (*Figure 3B*), as previously reported (*Yu et al., 2014*) and no Annexin-V binding was observed.

There are two explanations for the lag between ANO6 current activation and Annexin-V binding. One possibility is that current is required for PLS. However, we believe that the lag is partly explained by inherent differences in way the two events are measured. Patch clamp recording measures membrane conductance instantaneously, while detection of Annexin-V fluorescence requires the accumulation of Annexin-V on the membrane that is limited by its binding kinetics and the sensitivity

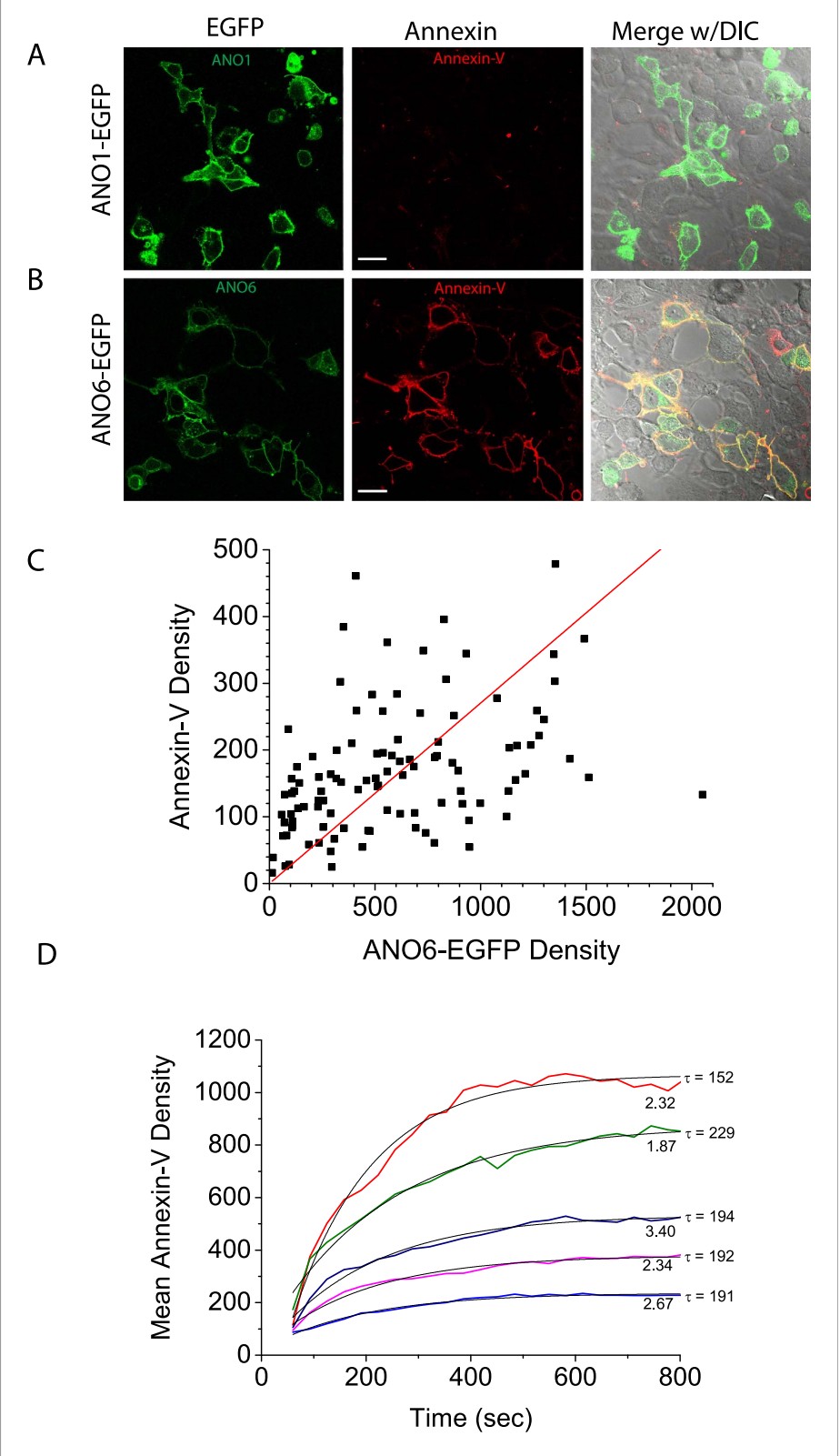

**Figure 2**. Characteristics of PLS linked to ANO6. Intracellular Ca$^{2+}$ was elevated by A23187, as in *Figure 1*, in polyclonal lines of HEK cells expressing (**A**) ANO1-EGFP or (**B**) ANO6-EGFP. (**C**) The relationship between ANO6-EGFP expression (mean EGFP pixel density of each cell) and AnnexinV-Alexa568 binding (mean pixel density in the Annexin-V channel masked by the EGFP channel). The line is the best fit to a straight line with Pearson's

*Figure 2. continued on next page*

*Figure 2. Continued*

correlation coefficient r = 0.84. (**D**) Examples of the time course of Annexin-V binding to five individual cells in a typical experiment. The numbers at the end of the trace represent the relative EGFP density of the cell and τ is the time constant of a mono-exponential fit (light black line) of the data.

of the fluorescence detection. Because Annexin binding to model bilayers is known to be slow and requires a threshold PtdSer concentration (*Kastl et al., 2002*; *Shi et al., 2006*) and the numerical aperture of the microscope objective was 0.6, we believe that currents and PLS occur contemporaneously. This correlation suggested the possibility that currents reflect transmembrane ion leakage associated with the process of phospholipid transport. The alternative hypothesis that ionic current through ANO6 somehow activates PLS is excluded by the finding that PLS occurs normally in ANO6-expressing cells under conditions where there is no ionic current (e.g., cells voltage-clamped at 0 mV with identical intracellular and extracellular solutions where $E_{rev}$ for every ion is 0 mV).

## ANO6 current and PLS require the same $Ca^{2+}$ concentration for activation

If ANO6 current is a consequence of PLS and not a separate function of the protein, we would expect that current and PLS would require the same $Ca^{2+}$ concentration for activation. Activation of ANO6 current requires >20 μM free $Ca_i^{2+}$ and requires minutes to develop (*Figure 4A,B*). With 20 μM $Ca^{2+}$, neither PLS nor currents are observed even after 20 min of recording. Currents and PLS are consistently observed only with 200 μM $Ca^{2+}$. Although this finding does not exclude the possibility that ion conductance and PLS are separate functions of ANO6, it is consistent with the two functions being linked.

## The ANO6 current is non-selective

If one accepts the proposal that ANO6 currents and Annexin-V binding occur simultaneously, this suggests that ANO6 currents may represent the flux of ions through micro-disruptions of the lipid membrane occurring during PLS rather than ions flowing through a defined aqueous pore defined by ANO6 protein. If ANO6 currents are a consequence of PLS, we would predict that their ionic selectivity would be very low. To explore the idea that ANO6 currents are essentially leak currents, we examined the ionic selectivity of the currents appearing after PLS was activated. In comparison to ANO1 currents, which exhibit robust anion:cation selectivity ($P_{Na}/P_{Cl} = 0.03$), the ANO6 current is highly non-selective (*Figure 5*). The ionic selectivity sequence was $Na^+ > Cl^- > Cs^+ > NMDG^+$ ($P_{Na}/P_{Cl} = 1.38$, $P_{Cs}/P_{Cl} = 0.6$, $P_{NMDG}/P_{Cl} = 0.48$). These data are consistent with the permeation pathway of ANO6 being relatively large and capable of passing $NMDG^+$ which has a mean diameter of ~7.3 Å. The finding that ANO6 currents have very low ionic selectivity and are activated contemporaneously with PLS over the same $Ca^{2+}$ concentration range suggested that PLS and currents have the same underlying mechanism.

## Identification of a protein domain required for scrambling

Because ANO1 has no scramblase activity while ANO6 does (*Malvezzi et al., 2013*; *Terashima et al., 2013*; *Suzuki et al., 2013b*; *Brunner et al., 2014*), we hypothesized that ANO6 contains a domain responsible for PLS that is absent in ANO1. We employed computational approaches to gain insights into sequence differences that could define this functional difference. We analyzed Type-I and Type-II divergence between mammalian ANO1 and ANO6 as an indication of the functional relevance of different amino acids (*Gu, 2006*). Sequences used for the analysis are shown in *Figure 6—figure supplement 1* and an alignment of ANO6 and ANO1 is shown in *Figure 6—figure supplement 2*. Type I divergence occurs shortly after gene duplication and is characterized by amino acids that are highly conserved in one paralogous group of proteins and highly divergent in the other. Type II divergence occurs later when specific functions undergo positive selection within a paralogous group, resulting in conserved changes in amino acid properties. Type II divergence is exemplified by alignment positions that are identical within paralogous groups but have amino acids with radically different properties between paralogous groups. There are three major

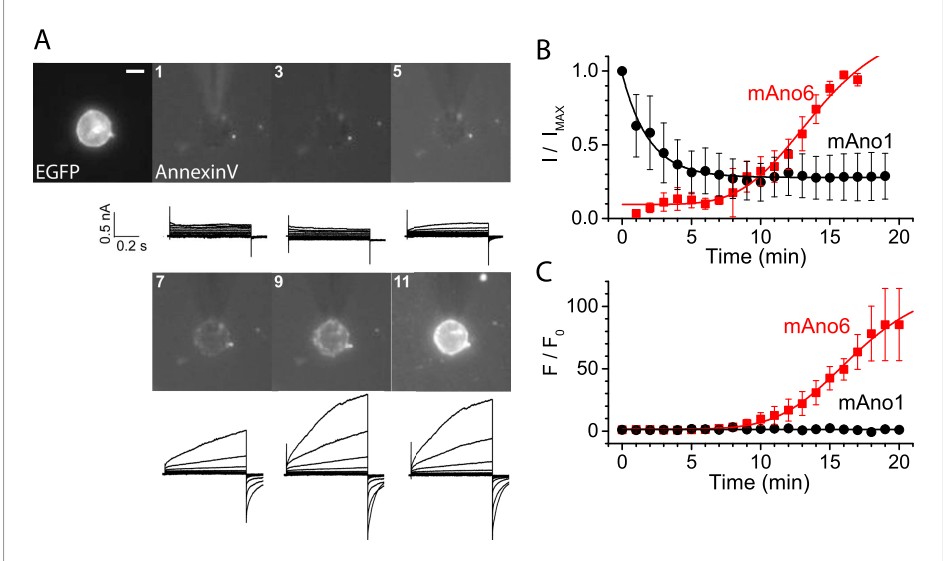

**Figure 3**. ANO6 current activates coincidently with PLS. HEK cells transiently expressing ANO6-EGFP were patch clamped in the presence of Annexin-V-Alexa-568 in the bath. The EGFP fluorescence image was obtained before establishing whole-cell recording and $F_o$ was determined immediately after establishing whole-cell recording. Annexin-V fluorescence images were acquired immediately after obtaining each I-V curve by voltage clamp at 1 min intervals. The patch pipet contained 200 µM free $Ca^{2+}$. I-V curves were obtained by voltage steps from −100 mV to +100 mV in 20 mV increments. (**A**) Representative images and currents of one of 20 experiments. The first image shows ANO6-EGFP fluorescence. The patch pipet can be seen entering the field from 12 o'clock. Scale bar 10 µm. (**B**) Average current amplitudes normalized to maximum current for cells expressing ANO6 (red square) or ANO1 (black circles) plotted vs time after establishing whole-cell recording. (**C**) Average Annexin-V fluorescence normalized to maximum fluorescence for the same cells as in **B** (n = 6).

regions of Type-II divergence between ANO1 and ANO 6 (**Figure 6A**). These regions are located in (a) intracellular loop 1, (b) TMD4 and TMD5 and the short intracellular loop between them, and (c) the C-terminus adjacent to the last transmembrane domain. To test the functional significance of these divergent amino acids, we made chimeric constructs of ANO1 and ANO6, named X-Y-X_*i-j*, where ANO paralog X has its amino acids *i-j* replaced with aligned amino acids from ANO paralog Y. The 1-6-1 chimeras, made by replacing short segments of ANO1 sequence with ANO6 sequence, were first screened by confocal microscopy of cultures.

Of 26 1-6-1 chimeras, 17 trafficked to the plasma membrane and generated Cl⁻ currents in patch clamp (**Figure 6B**, **Figure 6—figure supplement 3**). 13 1-6-1 chimeras did not exhibit PLS. However, four chimeras having ANO1 sequence replaced with ANO6 sequence in the region spanning TMD4 and TMD5 showed robust PLS activity (chimeras 1-6-1_D554-K588, 1-6-1_C559-F584, 1-6-1_S532-G558, and 1-6-1_D554-V569). The 1-6-1 chimera that scrambled having the smallest ANO6 sequence (1-6-1_D554-V569) had 15 amino acids of ANO1 replaced with amino acids 525–540 from ANO6. Additional constructs were made in which only pairs or triplet amino acids in ANO1 were mutated to the divergent amino acids from ANO6, but none of these chimeras exhibited PLS activity (**Figure 6—figure supplement 4**). We term this region of ANO6 between amino acids 525–559, capable of conferring PLS activity on ANO1, the scrambling domain (SCRD) (**Figure 6C**).

The ability of the SCRD to confer PLS activity on ANO1 does not necessarily prove that it is required for ANO6 PLS. To test this possibility, we mutated these amino acids in ANO6 to determine if they abolished ANO6-mediated PLS. Chimeras in which portions of the ANO6 SCRD were replaced with ANO1 sequence had greatly reduced PLS activity (**Figure 6—figure supplement 5**). Furthermore, mutation of several triplets of amino acids (525NTI, 529EKV, and 533IMI) between N525 and I535 abrogated ANO6 PLS activity.

Chimeras that trafficked to the plasma membrane, as judged by confocal microscopy, were then subjected to patch clamp analysis to measure ionic currents generated during PLS. **Figure 7A,B** shows

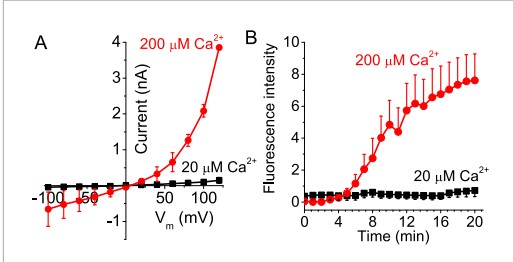

**Figure 4**. Activation of ANO6 current and PLS requires high intracellular $Ca^{2+}$ concentrations. (**A**) Average current–voltage relationships of currents recorded ∼20 min after establishing whole-cell recording in Ano6-expressing cells patched with 20 μM (black squares, N = 6) or 200 μM $Ca^{2+}$ (red circles, N = 10) in the patch pipet. (**B**) Annexin-V binding in Ano6-expressing cells patched with 20 μM (black squares, N = 5) and 200 μM (red circles, N = 15) $Ca^{2+}$ in the patch pipet. Error bars are S.E.M.

the percentage of cells that bound Annexin-V after 15 min whole cell patch clamp recording using 200 μM $Ca^{2+}$ in the patch pipet and the amplitude of the ionic current at +100 mV when Annexin-V binding had plateaued (see also *Figure 6—figure supplements 3–5*). Chimeras that support PLS invariably had ionic currents, while 6-1-6 chimeras that do not scramble (6-1-6_N525-Q559, 6-1-6_N525-I527, 6-1-6_E529-V531, and 6-1-6_N541-T662) exhibited very small or no currents. However, the amplitudes of the currents did not correlate with the percentage of cells that exhibited PLS, possibly because of differences in surface expression, which were not controlled (*Figure 7A,B*). The finding that mutations in ANO6 that abolished PLS also abolish ionic currents suggests that ions and phospholipids are translocated by overlapping molecular machinery.

To investigate the properties of the chimeras in more detail, we selected the PLS-positive chimera 1-6-1_D554-K588 and the PLS-negative chimera 6-1-6_N525-I527 for analysis. We first examined PLS in intact cells after elevation of cytosolic $Ca^{2+}$ with A23187. Like ANO6, 1-6-1_D554-K588 exhibited robust Annexin-V binding within 10 min (*Figure 7C*). In contrast, 6-1-6_N525-I527, which had three amino acids of ANO6 swapped with ANO1 sequence, did not exhibit PLS (*Figure 7C*). 96% of cells transfected with 1-6-1_D554-K588 (N = 178) of cells bound Annexin-V (*Figure 7D*). In contrast, only 15% of the cells (N = 127) transfected with 6-1-6_N525-I527 bound Annexin-V. The rate of Annexin-V binding was very similar for 1-6-1_D554-K588 and ANO6 (*Figure 7E*). On average, Annexin-V binding to ANO6-expressing cells occurred with τ = 143 s (*Figure 2D*), whereas Annexin-V binding in cells transfected with 1-6-1_D554-K588 occurred with τ = 218 s($χ^2$ = 0.03). Although other 1-6-1 chimeras that exhibited PLS were not investigated at the same detail, the characteristics of PLS of these chimeras were qualitatively similar to 1-6-1_D554-K588.

Patch clamp analysis shows that the currents of chimeras exhibiting PLS have properties that are a hybrid of ANO1 and ANO6 (*Figure 7—figure supplement 1*). While no PLS and no currents were observed in cells transfected with 6-1-6_N525-I527 during 20 min using 200 μM $Ca_i^{2+}$, in cells transfected with 1-6-1_D554-K588, robust Annexin-V binding occurred even with 20 μM $Ca_i^{2+}$. Therefore, it seems that this chimera retains the $Ca^{2+}$ sensitivity of ANO1. Moreover, the current was largest immediately upon establishing whole cell recording and then it ran down with time, similar to ANO1 currents (*Figure 7—figure supplement 1*). However, the rundown was not monotonic, but was interrupted by a transient increase in current. We interpret these data to indicate that this chimera initially exhibits ANO1-like currents, and that as PLS begins, the current acquires the properties of ANO6. The nature of this current is investigated below.

## Ionic currents associated with scrambling

We reasoned that if the pathway taken by ions is the same as the one taken by phospholipids, the two events might have similar pharmacology. Therefore, we tested the effects of MONNA, a small molecule that blocks ANO1 currents (*Oh et al., 2013*). We find that although 10 μM MONNA blocks ANO1 currents ∼90%, ANO6 currents are unaffected (*Figure 8A*). These data support the suggestion that the permeation pathway for $Cl^-$ in ANO1 differs from the ion conduction pathway of ANO6. Furthermore, MONNA does not block currents of 1-6-1 chimeras that exhibit PLS activity (*Figure 8A*). This suggests that the structural determinants of PLS in the SCRD alter that pharmacology of ANO1.

We have previously shown that ANO1 currents can be activated by strong depolarizations in the absence of $Ca^{2+}$ (*Xiao et al., 2011*). Because ANO6 does not exhibit such $Ca^{2+}$-independent currents, we asked whether we could detect $Ca^{2+}$-independent ANO1-like currents in 1-6-1_D554-K588 in zero $Ca^{2+}$.

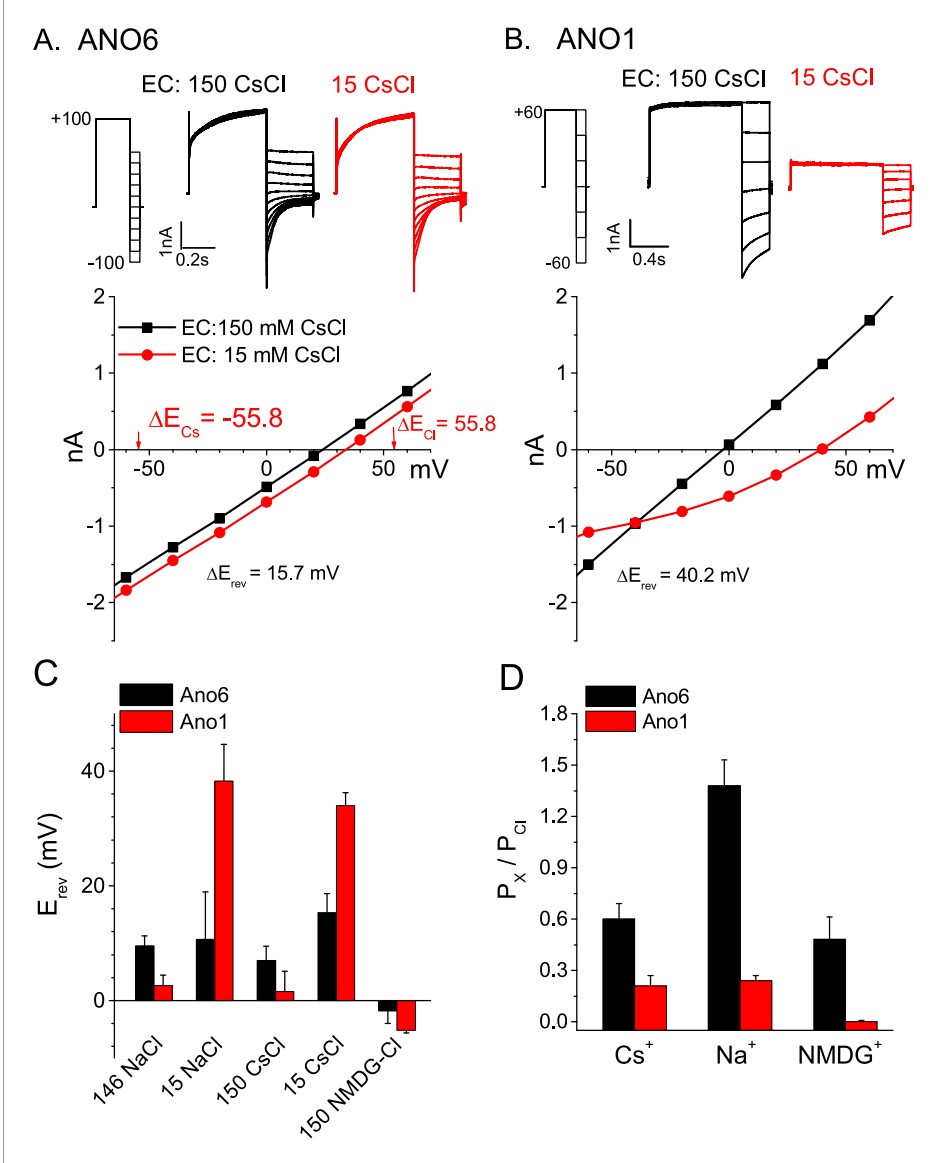

**Figure 5**. Ionic selectivity of ANO6 currents. Representative whole-cell patch-clamp recordings and current–voltage relationships from (**A**) ANO6 and (**B**) ANO1 expressing cells with 200 μM $[Ca^{2+}]_i$. Currents were recorded in 150 mM or 15 mM extracellular CsCl. The reversal potentials ($E_{rev}$) shift very little with ANO6-expressing cells, while the shift is large for ANO1-expressing cells. (**C**) Average $E_{rev}$ values for ANO6 or ANO1 expressing cells bathed in 146 NaCl, 150 CsCl, 15 NaCl, 15 CsCl, or 150 NMDG-Cl. (**D**) Relative permeabilities calculated from the Goldman-Hodgkin-Katz equation. N = 6–17.

Like ANO1, 1-6-1_D554-K588 exhibits $Ca^{2+}$-independent currents at very positive potentials and these currents are blocked by MONNA (**Figure 8B**, blue triangles). This current is $Cl^-$-selective ($P_{Cs}$:$P_{Cl}$ = 0.08) (**Figure 8C**). Thus, in zero $Ca^{2+}$ 1-6-1_D554-K588 currents resemble ANO1. However, after PLS has been activated by $Ca^{2+}$, the current is not blocked by MONNA (**Figure 8B**, red squares) and the current is >fourfold less $Cl^-$-selective ($P_{Cs}$:$P_{Cl}$ = 0.39) (**Figure 8D**). These data suggest that as PLS develops in response to elevated intracellular $Ca^{2+}$, the ionic conductance pathway changes from $Cl^-$-selective to non-selective. In fact, this transition between MONNA-sensitive and MONNA-insensitive currents can be seen when MONNA is applied shortly after establishing whole-cell recording with $Ca^{2+}$ in the pipet (**Figure 8B**, black open circles). The current is initially blocked partially by MONNA, but then the current begins to increase coincidently with development of PLS.

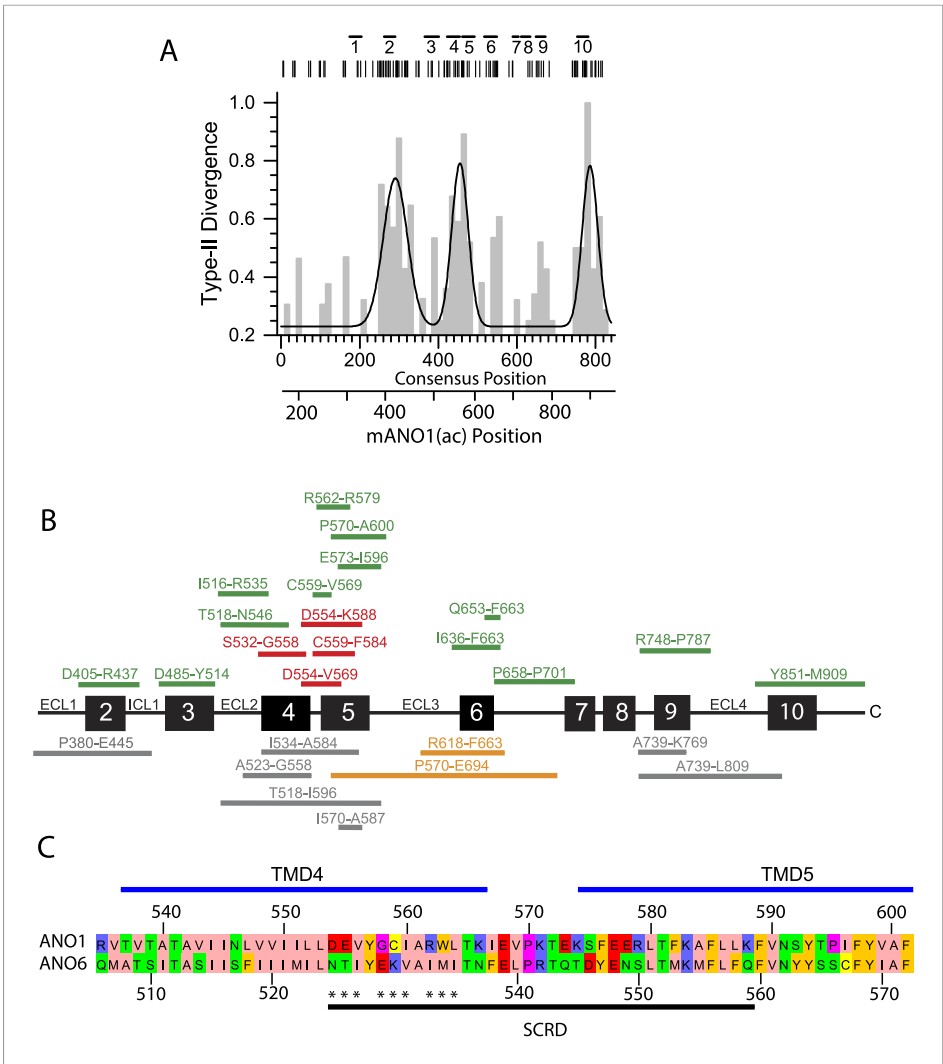

**Figure 6**. Identification of a PLS domain in ANO6. (**A**) Site-specific Type II divergence scores were calculated by comparing ANO1 and ANO6 sequences from 24 mammalian species, binned over a 15-amino acid window, and normalized to the maximum value. Horizontal lines at top indicate transmembrane domains. Vertical lines indicate individual amino acids with high Type II Divergence Scores. (**B**) Summary of 1-6-1 chimeras. Black line represents ANO1 sequence with TMDs 2–10 labeled. Colored horizontal lines represent ANO1 sequence that was replaced with the corresponding ANO6 sequence. Numbers refer to ANO1 sequence. Green: ANO1-like currents, no scrambling. Red: scrambling. Grey: weak plasma membrane expression, but no currents or scrambling. Orange: some plasma membrane expression, no currents or scrambling. (**C**) Scrambling domain (SCRD). The sequences of ANO1 (535–601) and ANO6 (506–572) are aligned with TMD 4 and TMD5 indicated. Amino acids are colored according to Rasmol. SCRD shows region associated with PLS. Asterisks show amino acids that are essential for PLS in ANO6. *Figure 6—figure supplement 1* lists sequence accession numbers used for Diverge analysis. *Figure 6—figure supplement 2* shows alignment of mANO1 and mANO6 used for chimeric construction. *Figure 6—figure supplement 3* tabulates the properties of all of the 1-6-1 chimeras. *Figure 6—figure supplement 4* shows a confocal image of scrambling by the 1-6-1_(554-588) chimera. *Figure 6—figure supplement 5* summarizes properties of mutations in ANO6 SCRD.

The following figure supplements are available for figure 6:

**Figure supplement 1**. Genbank accession numbers of sequences of mammalian species ANO1 and ANO6 used for DIVERGE analysis.

**Figure supplement 2**. MUSCLE alignment (*McWilliam et al., 2013*) of mANO1(ac) and mANO6 used for constructing chimeras.

*Figure 6. continued on next page*

*Figure 6. Continued*

**Figure supplement 3**. Properties of 1-6-1 chimeras that trafficked to the plasma membrane and generated ionic currents.
**Figure supplement 4**. Properties of 1-6-1 chimeras in which pairs or triplets of amino acids were mutated.
**Figure supplement 5**. Properties of ANO6 with mutations in the SCRD.

---

Because PLS of 1-6-1_D554-K588 develops quickly after establishing whole cell recording, it is difficult to compare the properties of the currents before and after PLS in the same cell. Therefore, we used the chimera 1-6-1_S532-G558 which activates PLS more slowly. About 5 min after establishing whole-cell recording before PLS was detectable, the current exhibited Cl$^-$-selectivity as indicated by a 35 mV shift in E$_{rev}$ upon switching from 150 mM CsCl to 15 CsCl ($P_{Cs}$:$P_{Cl}$ = 0.15) (*Figure 8E*). In contrast, after scrambling had occurred, cation permeability was significantly increased, with an $\Delta E_{rev}$ shift of only 15 mV ($P_{Cs}$:$P_{Cl}$ = 0.50) (*Figure 8F*). We cannot formally determine whether the Cl$^-$-selective pathway in the absence of PLS and the non-selective PLS-linked pathway are the same, but the observation that MONNA blocks the Cl$^-$-selective pathway but has no effect on the current after PLS develops suggests that the Cl$^-$-selective pathway is no longer present after PLS occurs. This is consistent with the Cl$^-$-selective pathway becoming less selective during scrambling.

## Homology model of ANO6

Recently, the X-ray structure of an ANO homolog from the fungus *Nectria haematococca* was published (*Brunner et al., 2014*). This protein, nhTMEM16, exhibits PLS activity when reconstituted into liposomes. Evidence for its channel activity is lacking, although the authors indicate that the lack of channel activity might be an artifact of the purification and reconstitution conditions. To gain insights into the structure-function relationships of the SCRD to protein structure, we created a homology model of ANO6 based on nhTMEM16 (*Figure 9*). This model shows that the SCRD forms a unique structure. Along with helices 3, 6, and 7, it forms what Brunner et al. (*Brunner et al., 2014*) call the 'subunit cavity', a hydrophilic crevice that faces the lipid membrane on the side of the protein away from the dimer interface. Brunner et al. speculate that this region may be involved in lipid transport.

## Discussion

### The scrambling pathway of ANO6

PLSases are conceived to function in a fashion similar to ion channels by forming an aqueous pore for the polar head groups of the phospholipids to cross the hydrophobic center of the membrane (*Pomorski and Menon, 2006*; *Sanyal and Menon, 2009*). The recent 3-dimensional structure of nhTMEM16, which catalyzes PLS when incorporated into lipid vesicles, provides key insights into this process (*Brunner et al., 2014*). The authors noticed a peculiar hydrophilic cavity facing the hydrophobic bilayer that they propose may be involved in lipid scrambling, although no lipids were visualized in the structure. The idea that this cleft may be the phospholipid 'channel' is supported by our ANO6 homology model that predicts that the SCRD forms one side of this cleft. Intriguingly, residues located in TMD7-TMD8 of ANO1, identified for their roles in Ca$^{2+}$ binding and gating, make up the other side of this hydrophilic cavity (*Yu et al., 2012*; *Tien et al., 2014*; *Bill et al., 2015*) (*Figure 9D,E*). The Ca$^{2+}$ binding site, therefore, is well situated to control structural rearrangements of the hydrophilic cleft to provide a potential pathway for phospholipid translocation.

### Is ANO6 a PLSase?

An important role of ANO6 in PLS seems well established (*Suzuki et al., 2010*, *2013b*), although its identity as a PLSase has been contested (*Yang et al., 2012*). This debate has attracted considerable attention (*Harper and Poole, 2013*; *Kmit et al., 2013*; *Kunzelmann et al., 2014*; *Pedemonte and Galietta, 2014*). It is important to note that despite the motivation to conclude that ANO6 is a PLSase

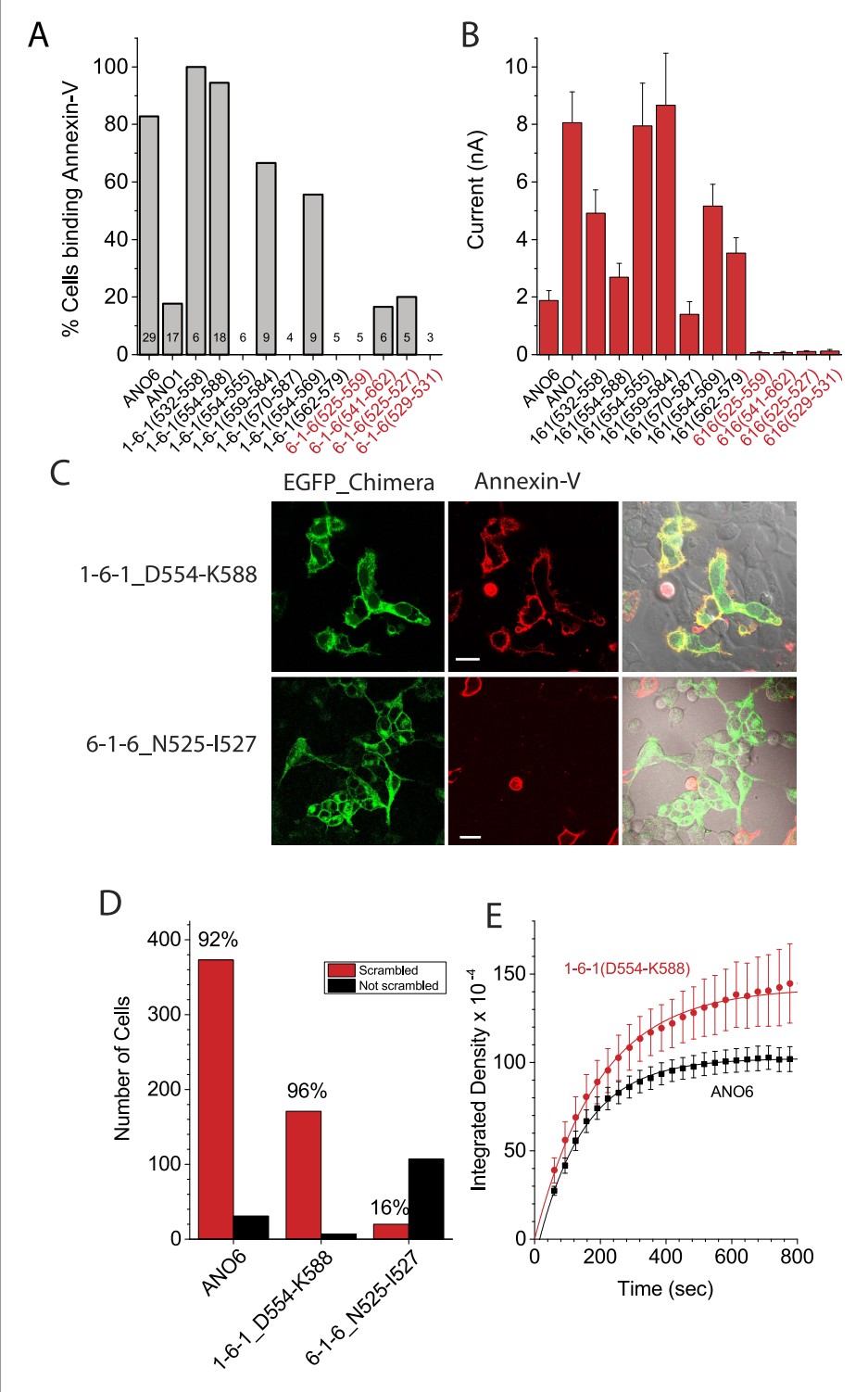

**Figure 7**. Properties of chimeras of ANO1 and ANO6. (**A**, **B**) Patch clamp analysis of PLS (**A**) and ionic currents (**B**) in ANO1-ANO6 chimeras. Cells were patch clamped with 200 μM Ca²⁺ in the pipet and PLS was monitored by Annexin-V binding and currents measured by voltage steps from 0 mV to +100 mV. (**C**) Confocal imaging of Annexin-V binding to HEK cells transfected with the 1-6-1_D554-K588 or the 6-1-6_N525-I527 chimeras 10 min after elevating Ca²⁺ with A231187. (**D**) Number of cells binding Annexin-V ('scrambled') or not binding Annexin-V ('not scrambled') 10 min after elevation of Ca²⁺ with A23187 (N = 3 experiments each). (**E**) Time

*Figure 7. continued on next page*

*Figure 7. Continued*

course of Annexin-V binding to cells expressing the 1-6-1_D554-K588 chimera (red) compared to the time course of Annexin binding to ANO1-expressing cells (black).

The following figure supplement is available for figure 7:

**Figure supplement 1**. Patch clamp analysis of ANO1-ANO6 chimeras.

based on its similarity in structure to nhTMEM16, it remains a possibility that ANO6 is a regulator of another protein that is the PLSase. Our finding that only about 92% of cells expressing ANO6 exhibit PLS suggests that ANO6 is not sufficient for PLS. Further, the lack of correlation between ANO6 expression levels and PLS activity supports some skepticism. One must only recall that PLSCR1 incorporated into liposomes mediates PLS but that knockout of its gene does not have clear-cut effects on PLS to appreciate a need for caution (*Acharya et al., 2006*; *Fadeel and Xue, 2009*; *Bevers and Williamson, 2010*; *Ory et al., 2013*).

## The ion conduction pathway

We propose that during scrambling ions pass through the same pathway that is occupied by the phospholipid. Ions might accompany the phospholipids as counterions or may simply flux independently through the same pathway. This suggestion is based on the observation that PLS and ionic currents activate contemporaneously and exhibit similar sensitivity to $Ca^{2+}$. Furthermore, the presence of non-selective, slowly-activating currents that are insensitive to the ANO1 inhibitor MONNA correlate with the ability of chimeras to scramble. Some support for this idea is provided by cysteine mutagenesis and accessibility experiments showing that amino acids that we believe form the vestibule of the $Cl^-$ selective pore of ANO1 (*Yu et al., 2012*) are located at the extracellular surface of the hydrophilic cleft (*Figure 9—figure supplement 1*).

## Evolution of the ANO/TMEM16 family

Our data add to the already growing awareness that the ANO family is functionally split with some of its members being anion-selective ion channels (ANO1 and ANO2) and other members having the ability to transport lipids between membrane leaflets. This functional duplicity is reminiscent of two other anion channels, CFTR and CLCs, which apparently evolved from transporters. CFTR (cystic fibrosis transmembrane conductance regulator) is a $Cl^-$ channel that evolved from ABC transporters (*Gadsby et al., 2006*; *Jordan et al., 2008*; *Miller, 2010*), and the CLC chloride channels CLC-1 and CLC-2 are members of a 9-gene family most of which are $H^+$-$Cl^-$ exchangers (*Miller, 2006*; *Lisal and Maduke, 2008*). Additionally, the P4 ATPases that function as lipid flippases evolved from a family of ion transporters (*Baldridge and Graham, 2013*; *Lopez-Marques et al., 2014*). One might speculate that the primordial ANO was a lipid transporter. During evolution, functional divergence likely occurred after gene duplication where the selective pressure for the major function (lipid transport) decreased, thus enabling the enhancement of the minor sub-function (ion transport). The finding that fungi have only one ANO gene, but that some fungal ANO homologs have PLS activity (*Malvezzi et al., 2013*; *Brunner et al., 2014*), supports the idea that PLS is an ancient function of ANOs. Our data suggest that the ionic currents carried by ANO6 are essentially flowing along with the scrambling lipids. Thus, the non-selective ion transport that occurs concomitantly with PLSase activity may have been a sub-function in the ancestral ANO. After gene duplication occurred, this sub-function may have evolved $Cl^-$ ion specificity.

ANOs that have evolved into anion channels may be more $Cl^-$ selective than the scrambling ANOs simply because the energy contour of the conduction pathway is altered by the absence of lipid substrate. Because $Cl^-$ is more hydrophobic than its cationic cousins, it may be more suited to traverse the evolutionary remnants of the phospholipid pathway. Consistent with this idea is the fact that, compared to many voltage-gated cation channels, $Cl^-$ channels are sadly non-selective: virtually all anions permeate and cations are often significantly permeable (*Duran et al., 2010*). The pores of $Cl^-$ channels are often modeled as large viaducts that provide selectivity based largely on ionic hydration energies (*Dawson et al., 1999*; *Qu and Hartzell, 2000*; *Liu et al., 2003*). This low selectivity is

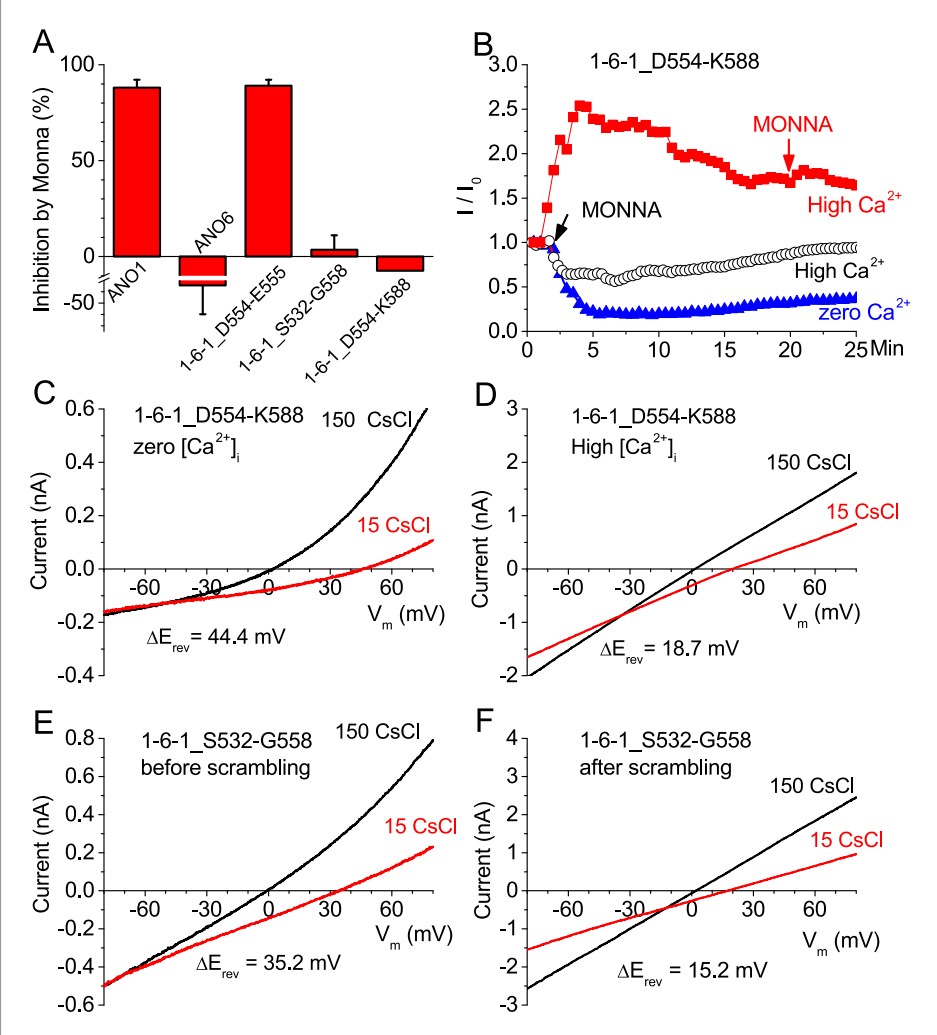

**Figure 8**. Ion channel properties of ANO1-ANO6 chimeras. (**A**) Inhibition of currents by MONNA. ANO1 is nearly completely blocked by 10 μM MONNA, while ANO6 and the 1-6-1 chimeras that scramble are not affected. (**B**) Effects of MONNA on currents associated with the 1-6-1_D554-K588 chimera. MONNA blocks the $Ca^{2+}$-independent current elicited by depolarization to +100 mV (blue triangles), but has no effect on the $Ca^{2+}$-activated current after scrambling has occurred (red squares). In contrast, before scrambling has occurred, MONNA partially blocks the $Ca^{2+}$-activated current (open circles). (**C–F**) The ionic selectivity of currents associated with the 1-6-1_D554-K588 chimera (**C–D**) and the 1-6-1_S532-G558 chimera (**E**, **F**) were determined by the dilution method (see 'Materials and methods') by measuring reversal potentials with external solutions containing either 150 mM (black line) or 15 mM (red line) CsCl. (**C**) $Ca^{2+}$-independent (zero $Ca_i^{2+}$) and (**D**) $Ca^{2+}$-activated currents associated with the 1-6-1_S532-G558 chimera. Currents recorded (**E**) before and (**F**) after scrambling with the 1-6-1_S532-G558 chimera. N = 3–7. Error bar represents SEM.

precisely what one might expect to evolve from a transporter that previously transported phospholipids. The finding that we can convert ANO1 into a protein capable of catalyzing PLS provides further support for this hypothesis for ANO1 evolution. Like the ClC family, where a single glutamic acid residue can determine whether a protein is a $H^+$-$Cl^-$ transporter or a $Cl^-$ channel, here we show that a few amino acids can convert ANO1 into a protein that supports PLS activity. However, we have not yet been able to convert ANO6 into a $Cl^-$ selective ion channel. It is likely that additional changes outside the SCRD are required to confer $Cl^-$ selectivity. This may be also be explained by epistasis, where evolutionary trajectories are blocked by the accumulation of neutral mutations that have no impact on the initial function (non-selective ion transport) but prevent acquisition of the new function ($Cl^-$ selectivity) (*Bridgham et al., 2009*; *Breen et al., 2012*). *Yang et al. (2011)* have reported that mutations of the last

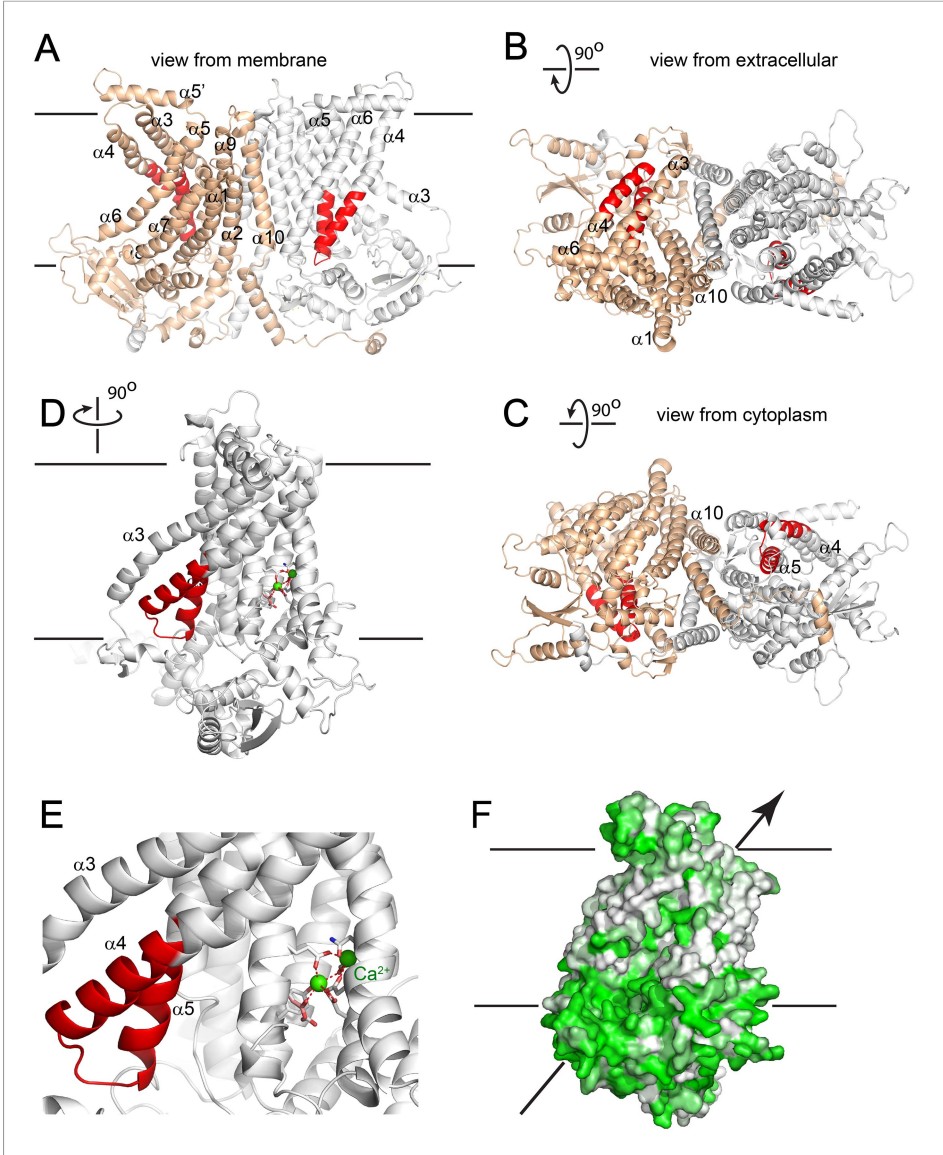

**Figure 9**. Homology model of ANO6. (**A**) Side view from the membrane. ANO6 is shown as a dimer with the left subunit in gold and the right subunit in grey. The SCRD is colored red. Transmembrane helices are numbered. (**B**) View from extracellular side. (**C**) View from cytoplasm. (**D**) A view from the membrane looking towards the hydrophilic cleft showing the SCRD in red and the $Ca^{2+}$ binding site in stick representation. Residues conserved between ANO6 and TMEM16 that coordinate $Ca^{2+}$ (green) are shown as white sticks (C = white, O = red). ANO6 contains GXXX (pink), which is a D503 in TMEM16. (**E**) Close-up view of **D**. (**F**) Same view as **D** with surface colored to show hydrophilicity. Green: hydrophilic. White: hydrophobic.

The following figure supplement is available for figure 9:

**Figure supplement 1**. Homology model of ANO1 dimer.

amino acid in the SCRD (Q559 in ANO6, K588 in ANO1) alters the ionic selectivity, but the changes are only 2–3-fold and do not shift selectivity from cation to anion.

Intriguingly, the *Saccharomyces* ANO homolog, Ist2p, has been shown to play a role in tethering the cortical ER to the plasma membrane, but it is not known whether this protein also has PLSase or ion channel activity. However, one might speculate wildly that Ist2p may play a role in transport of lipids from the ER to the plasma membrane, because tethering of the ER to the plasma membrane is

important for the proper function of non-vesicular lipid transport (*Holthuis and Menon, 2014*) and deletion of Ist2p along with other ER-plasma membrane tethering proteins results in aberrant phosphatidylinositol 4-phosphate levels and localization (*Stefan et al., 2011*; *Manford et al., 2012*; *Wolf et al., 2012*; *Stefan et al., 2013*).

## Is ANO1 a PLSase?

ANO1 is well established as the pore forming unit of $Ca^{2+}$ activated $Cl^-$ channels (*Caputo et al., 2008*; *Schroeder et al., 2008*; *Yang et al., 2008*) and is incapable of rescuing PLS in cells with ANO6 disrupted (*Suzuki et al., 2013b*). However, our finding that the function of ANO1 can be converted by replacing a domain as small as 15 amino acids raises questions whether ANO1 might be a PLSase under appropriate conditions. Perhaps a missing subunit or regulatory enzyme could activate its PLSase activity. Alternatively, the SCRD may hold a vital interaction site for another component essential for this process or possess a key site of posttranscriptional modification required for activating this activity. In any case, it is clear that ANO1 function is intimately dependent on phospholipids (*Terashima et al., 2013*) and understanding the relationships of the ANOs to membrane lipids is certain to be a major research goal for many laboratories in the coming years.

# Materials and methods

## cDNAs

mANO1 (Uniprot Q8BHY3) and mANO6 (Uniprot Q6P9J9) tagged on the C-terminus with EGFP were provided by Dr Uhtaek Oh, Seoul National University. For designing chimeras, mANO1 and mANO6 were aligned using MUSCLE (*McWilliam et al., 2013*). Chimeras were constructed using overlap extension PCR (*Pont-Kingdon, 1997*). Chimeras are named X-Y-X_*i-j*, where X is the ANO paralog template whose amino acids numbered *i-j* are replaced with the aligned amino acids from ANO paralog Y. The alignment is shown in *Figure 6—figure supplement 2*. PCR primers were designed to engineer complementary overlapping sequences onto the junction-forming ends of PCR products that were subsequently assembled by PCR. PCR-based mutagenesis was used to generate mutations in one or a few amino acids. The protein coding region of all chimeras and mutants were sequenced.

## Clover-(His)$_6$-LactC2

A cDNA construct consisting of Clover fluorescent protein followed by a hexa-histidine tag and lactadherin-C2 (Clover-(His)$_6$-LactC2) in the pET-28 bacterial expression vector was a generous gift from Dr Leonid Chernomordik (NIH/NICHD). Rosetta2(DE3) BL21 *Escherichia coli* (Novagen, Germany) were transformed with Clover-(His)$_6$-LactC2 in pET28, grown in TB medium at 37°C in 50 µg/ml kanamycin and 30 µg/ml chloramphenicol until the culture reached $A_{600} = 1$. After addition of 1 mM IPTG, the culture was grown for 3 hr at 28°C. Cells were lysed in B-Per (Thermo Scientific, MA) containing lysozyme, benzoase, and protease inhibitor cocktail III (Calbiochem). After centrifugation at 20,000×*g* 10 min, Clover-(His)$_6$-LactC2 was purified from the supernate on a 1-ml Talon cobalt affinity column (Clontech, CA). Stock solutions of 1–3 mg/ml were stored in the elution buffer (150 mM imidazole pH 7 plus 0.02% $NaN_3$) and were used at ~1–3 µg/ml.

## Patch clamp electrophysiology

HEK293 cells were transfected with ~1 µg cDNA per 35 mm dish using Fugene-9 (Roche Molecular Biochemicals, Indianapolis, IN). Single cells identified by EGFP fluorescence were patch clamped ~2 days after transfection. Transfected cells were identified on a Zeiss Axiovert microscope by EGFP fluorescence. Cells were voltage-clamped using conventional whole-cell patch-clamp techniques with an EPC-7 amplifier (HEKA, Germany). Fire-polished borosilicate glass patch pipettes were 3–5 MΩ. Experiments were conducted at ambient temperature (22–26°C). Because liquid junction potentials calculated using pClamp were predicted to be <2 mV, no correction was made. The zero $Ca^{2+}$ pipet (intracellular) solution contained (mM): 146 CsCl, 2 $MgCl_2$, 5 EGTA, 10 sucrose, 10 HEPES pH 7.3, adjusted with NMDG. The ~20 µM $Ca^{2+}$ pipet solution contained 5 mM $Ca^{2+}$-EGTA. The 0.2 mM $Ca^{2+}$ solution was made by adding additional 0.2 mM $CaCl_2$ to the 20 µM $Ca^{2+}$ pipet solution. The standard extracellular solution contained (mM): 140 NaCl, 5 KCl, 2 $CaCl_2$, 1 $MgCl_2$, 15 glucose, 10 HEPES pH 7.4. For determination of ionic selectivity, the external solution contained 2 $CaCl_2$, 1 $MgCl_2$, 15 glucose,

10 HEPES pH 7.4, and various concentrations of NaCl, CsCl, or NMDG-Cl as indicated. The internal solution contained (mM): 150 NaCl (or CsCl), 1 MgCl$_2$, 5 Ca-EGTA plus 0.2 CaCl$_2$, and 10 HEPES pH 7.4. The osmolarity of each solution was adjusted to 300 mOsm by addition of mannitol. Relative permeabilities of cations (X) relative to Cl$^-$ were determined by measuring the changes in zero-current $E_{rev}$ when the concentration of extracellular NaCl or CsCl was changed ('dilution potential' method) as previously described (*Barry, 2006*; *Yu et al., 2012*). Relative permeabilities were calculated from the shift in $E_{rev}$ calculated using the Goldman-Hodgkin-Katz equation:

$$\Delta E_{rev} = 25.7 \ln\left[(X_o + Cl_i * P_{Cl}/P_{Na}) / (X_i + Cl_o * P_{Cl}/P_{Na})\right],$$

where X is the cation and $\Delta E_{rev}$ is the difference between $E_{rev}$ with the test solution XCl and that observed with symmetrical solutions.

## PLS

PLS was assessed by live-cell imaging of the binding of Annexin-V conjugated to Alexa Fluor-568 (Invitrogen, CA; 'Annexin-V' diluted 1:200) or LactC2 fused to Clover fluorescent protein (*Kay et al., 2012*). PLS was measured in populations of intact HEK293 cells grown on glass coverslips mounted in Attofluor chambers (Invitrogen, CA) and imaged at ambient temperature with a Zeiss confocal microscope using a 63× Plan-Aprochromat NA 1.4 objective. PtdSer exposure was measured by binding of Annexin-V-AlexaFluor-568 (543 nm excitation; 560 nm long pass emission) or LactC2-Clover (488 nm excitation, 500–530 nm band pass emission). The bath solution contained (mM) 140 NaCl, 10 CaCl$_2$, 10 Tris-HCl pH 7.4. PLS was stimulated by elevation of intracellular Ca$^{2+}$ using the Ca$^{2+}$ ionophore A23187 (10 μM). The most reproducible PLS was obtained by incubation of cells 5 min in A23187 in nominally zero-Ca$^{2+}$ solution followed by washout of A23187 and addition of 5 mM Ca$^{2+}$ to initiate PLS. We presume that the A23187 exposure in zero Ca$^{2+}$ depletes internal stores and that the readdition of Ca$^{2+}$ results in store-operated Ca$^{2+}$ entry that is more rapid than Ca$^{2+}$ entry through A23187 channels alone. In some experiments, PLS was stimulated by exposure to A23187 and Ca$^{2+}$ simultaneously Sometimes SERCA inhibitors were included with no obvious effect. Scrambling was quantified by measuring the increase in Annexin-V fluorescence in EGFP-expressing cells by creating a binary mask from the EGFP channel for each Annexin-V-Alexa-568 frame in the time series using Fiji Image-J 1.49. The raw integrated density was then calculated by adding the intensity of all pixels in the unmasked area.

Binding of Annexin-V-AlexaFluor-568 to patch-clamped cells during voltage-clamp recording was imaged with a wide-field Zeiss Axiovert 100 microscope using a 40× NA 0.6 LD-Acroplan objective. Images were acquired with an Orca-FLASH 4.0 digital CMOS camera (C11440, Hamamatsu, Japan) controlled by Metamorph 7.8 software (Molecular Devices, CA). Annexin-V-AlexaFluor-568 was added to the normal extracellular solution before patch clamping the cell. After whole-cell recording was established with an intracellular solution containing either zero, 20 μM, or 200 μM free Ca$^{2+}$ the accumulation of Annexin-V on the plasma membrane was imaged at 1-min intervals synchronously with voltage clamp recording.

### Immunostaining and antibodies

Stable ANO6-FLAG$_{3X}$ cells were fixed on glass coverslips in 4% PFA for 10 min at room temperature, permeablized with 1.5% Tritonx-100 and stained with anti-FLAG M2 antibody (Sigma–Aldrich, MO) for two hours at room temperature. Coverslips were washed and stained with anti-mouse-Alexa-488 secondary and Alexa-633 conjugated phalloidin (1:1000, Molecular Probes, CA). Western blot analysis was performed on protein lysates using anti-FLAG M2 antibody and anti-GAPDH (Millipore, MA) followed by incubation with HRP conjugated anti-mouse secondary (BioRad, CA).

### Divergence analysis

Type-II divergence was determined using DIVERGE 3.0 (http://xungulab.com/software.html) (*Gu, 2006*; *Gu et al., 2013*). Sequences used for the DIVERGE analysis are listed in Figure 5—figure supplement 1. Sequences were curated, divergent N-terminal and C-terminal sequences were deleted, and the sequences aligned using MUSCLE. For plotting, the site-specific posterior ratios for Type II divergence were binned across a window of 15 amino acids and normalized to the maximum value.

## Homology models

We created homology models of ANO6 based on nhTMEM16 (*Brunner et al., 2014*). mANO6 sequence was submitted to the Phyre2 Protein Fold Recognition Server (*Kelley and Sternberg, 2009*). mANO6 (truncated to 833 residues by deleting the extreme N- and C- termini) was aligned to the nhTMEM16 sequence extracted from PDB ID 4WIT (674 residues). These two sequences share 23% identity and 37.4% similarly (calculated by the BLSM62 algorithm). A total of 12 gaps were introduced, primarily at the N- and C-termini and in loops between secondary structural elements. Secondary structure prediction was incorporated into the alignment for Phyre2 (and all of the other modeling servers used for comparison including Swiss-Model and Tasser). There are no gaps in or near the SCRD, with the nearest gaps located in extracellular loops between helices 3 and 4 (2 residue gap 24 residues away) and between helices 4 and 5 (1 residue gap 22 residues away). While overall alignment and homology model generation would benefit from a template with greater sequence identity, all modeling servers reported high confidence in the generated models with no variation in the placement of the SCRD between models. Model geometry was minimized and the model subjected to 500 steps of geometry idealization and energy minimization using Phenix (*Adams et al., 2010*). The overall RMSD of Cα atoms in the final alignment was 1.485 Å with a Q-score of 0.667 (calculated by UCSF Chimera).

## Analysis of data

Electrophysiological traces were analyzed with Clampfit 9 (Molecular Devices, CA). Fluorescence intensity was analyzed with MetaMorph 7.8 and Fiji Image-J 1.49. Data are presented as mean $\pm$ SEM. Statistical difference between means was evaluated by two-tailed t-test. Statistical significance was assumed at $p < 0.05$.

## Acknowledgements

Supported by NIH grants GM60448-12 and EY114852-11 to HCH. Jarred Whitlock was supported by an NIH Training Grant 5T32GM008367-25. This research project was supported in part by the Emory University Integrated Cellular Imaging Microscopy Core.

## Additional information

### Funding

| Funder | Grant reference | Author |
| --- | --- | --- |
| National Institutes of Health (NIH) | GM60448-12 | H Criss Hartzell |
| National Institutes of Health (NIH) | EY114852-11 | H Criss Hartzell |
| National Institutes of Health (NIH) | 5T32GM008367-25 | Jarred M Whitlock |

The funder had no role in study design, data collection and interpretation, or the decision to submit the work for publication.

### Author contributions

KY, Conceived and designed experiments, Performed and analyzed all patch clamp experiments, Prepared figures, Wrote the manuscript; JMW, Contributed to experimental design, Performed and analyzed scrambling experiments, Helped write the manuscript; KL, Assisted with phospholipid scrambling experiments, Tissue culture, Helped to write and edit the manuscript; EAO, Assisted with development and analysis of homology models; YYC, Designed and made all the cDNA constructs, chimeras, and mutants, Maintained tissue culture, Performed transfections; HCH, Conceived and designed experiments, Performed and analyzed scrambling experiments, Edited figures, Wrote the manuscript

### Author ORCIDs

Jarred M Whitlock, http://orcid.org/0000-0002-5886-2047
Kyleen Lee, http://orcid.org/0000-0003-2418-226X
H Criss Hartzell, http://orcid.org/0000-0002-3393-1528

## Additional files

### Major dataset

The following previously published dataset was used:

| Author(s) | Year | Dataset title | Dataset ID and/or URL | Database, license, and accessibility information |
|---|---|---|---|---|
| Brunner JD, Lim NK, Schenck S, Duerst A, Dutzler R | 2014 | TMEM16 lipid scramblase in crystal form 2 | http://www.rcsb.org/pdb/explore/explore.do?structureId=4WIT | Publicly available at RCSB Protein Data Bank (Accession No. 4WIT). |

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
