## [Decision Letter]

Thank you for sending your work entitled “Identification of a lipid scrambling domain in ANO6/TMEM16F” for consideration at *eLife*. Your article has been favorably evaluated by Randy Schekman (Senior editor and Reviewing editor) and three outside reviewers.

The following individual responsible for the peer review of your submission have agreed to reveal their identity: Todd Graham and Anant Menon (peer reviewers). A further reviewer remains anonymous.

The Reviewing editor and the reviewers discussed their comments before we reached this decision, and the Reviewing editor has assembled the following comments to help you prepare a revised submission.

We agree that this work is important and for the most part a compelling case is made that ANO6 is indeed a phospholipid scramblase. This work certainly merits publication in *eLife* pending some additional work that is described in detail in the three referee reports appended below. However the reviewers have conferred and decided to focus your attention on four main points that must be satisfied to make this work publishable:

1) The authors should improve the annexin quantification and the correlation of its response with ANO6 expression.

2) The lag time between the ionic current and annexin binding can be explained by the non-linear response of annexin to PS density. A few experiments with Lactadherin, whose response to PS is more linear, could strengthen this part and validate the hypothesis that the ionic current is a consequence of the scramblase activity, not the other way round.

3) In the present state, the lipid docking predictions are not detailed enough to be convincing. The authors should provide more information or delete all this part.

4) In general, the paper would benefit from a major re-write to reduce jargon, and to state the problem and results clearly. Referencing would also benefit from a better appreciation of the lipid scrambling field.

There are also many other issues that you could address but the four above are the main ones that the reviewers consider essential.

Reviewer #1:

There are conflicting reports on whether ANO6/TMEM16F is a Ca^2+^-activated ion channel, a Ca^2+^-activated phospholipid scramblase, or both. There is also significant discrepancy in the literature on the ion selectivity of ANO6, and assuming it can directly scramble phospholipid, there is no clear understanding of how ANO6 mediates phospholipid transport while the closely related ANO1 clearly lacks scramblase activity. The submitted manuscript provides evidence that ANO6 primarily functions as a phospholipid scramblase and that ion permeation through the channel is a consequence of phospholipid translocation.

Part of the significance of this work is that it potentially clarifies the relationship between the ion channel and scramblase activities of ANO6. A second, major point of significance is that the authors help define the mechanism of phospholipid scrambling by mapping the region of the TM domain that is responsible for this activity. By transplanting a small section of the 4th and 5th transmembrane segments of ANO6 into ANO1, they conferred upon ANO1 the ability to induce Ca^2+^-dependent phospholipid scrambling in transfected cells. Within a homology model, this region maps to a hydrophilic cleft in the membrane domain that appears to be the ion permeation route in ANO1, but can be molded into a phospholipid transporter by exchange of a few amino acids. These observations provide the first mechanistic insight into how substrate specificity of the ANO/TMEM16 family of proteins is determined.

There are a few weaknesses in the presentation of the data that modestly detracts from my enthusiasm for the work.

Primary concerns:

1) A trivial explanation for the difference in scramblase activity for GFP-ANO1 and GFP-ANO6 shown in Figure 1 could be differences in the level of expression. Assuming the images were acquired with the same exposure settings, the authors could and should comment on the relative expression levels of their constructs.

2) For Figure 2, it seems odd to me that the transfected ANO1 or ANO6 confer the major currents in the plasma membrane of the HEK cells. What kind of currents do untransfected cells display in these patch clamp experiments? Have these currents been measured and subtracted from the transfected cell data? Phosphatidylserine (PS) is also carrying charge across the membrane and the authors should indicate whether or not this contributes to the current they are measuring.

3) There is a tremendous amount of data squeezed into Figure 5 and the impact/importance of the observations is somewhat obscured because of the complexity and density of information. In addition, the qualitative and more subjective nature of the “% scrambled cells” measure used in Figure 5 and “yes/no” for scrambling activity in the table was unsatisfying because it is unclear how well the 1-6-1 chimeras have recapitulated the Ca^2+^-activated scramblase activity of ANO6. My primary recommendation is that the authors take their best 1-6-1 chimera carrying the SCRD and their best 6-1-6 chimera which loses SCR activity and perform the more quantitative kinetic assay as shown in Figure 2. These data should be included in either Figure 5 or Figure 6.

4) The molecular docking of PS to the modeled structure in Figure 8 is premature and should be removed. I don't think anyone has addressed the phospholipid specificity of ANO6 and I wouldn't expect a strong and specific binding site for PS in the membrane domain that mediates scrambling. My suggestion to the authors is to simply display the homology model and delete the description of their attempts to dock PS. Nothing meaningful was gained by this approach and the description seemed to detract from the presentation. One doesn't need this result to speculate on the translocation pathway in the Discussion section. I would prefer to see a zoomed in view of the SCRD in relationship to the proposed Ca^2+^-binding site for Figure 8.

Reviewer #2:

The exposure of PtdSer on the external leaflet of the PM by phospholipid scrambling occurs in several biological processes and is an important cellular signaling mechanism. However, there is a great uncertainty about the identity of the phospholipid scramblases involved in such processes. This manuscript focuses on the lipid scrambling activity of ANO6/TMEM16F (hereafter ANO6), which has been proposed to be a Ca^2+^-activated scramblase. Yet, it has also been suggested that ANO6 is required for lipid scrambling by forming an ion channel but not by performing the scrambling activity itself. This manuscript convincingly shows that ANO6 is a ‘true’ scramblase and also indicates that the recorded ion channel activity of ANO6 is a consequence (a leak current) of phospholipid scrambling. In an elegant and very detailed manner the authors identify a minimal region in ANO6 required for PS scrambling. This is an excellent study.

1) Experiments include real-time monitoring of ion currents by patch clamp, and phospholipid scrambling using Annexin-V as indicator of exposed PtdSer. The authors find that ANO6 current is non-selective and allows passage of large ions such as NMDG+. The model concludes that ANO6 non-selective ion current coincides with phospholipid scrambling. Annexin-V has been widely used as a probe for exposed PtdSer, but Annexin-V response to PtdSer is not linear and requires a certain threshold of PtdSer to bind. Therefore, this probe is not well adapted for correlative real-time assays such as presented in Figure 2. Notably, Annexin-V signal appears 3 min after the increase in current amplitude. This lag is commented in the text and the hypothesis raised by the authors is reasonable. However, is it possible to do the same experiment using a more robust PtdSer probe such as Lactadherin? Advantages of Lactadherin over Annexin-V have been listed in Shi J et al. 2006, a paper also cited by the authors.

2) The authors performed extensive sequence swapping studies to define the so-called scrambling domain (SCRD) in ANO6, which can confer the scrambling activity to ANO1. Mutations in SCRD prevented ANO6 from performing PtdSer scrambling. The manuscript presents a homology model of ANO6 based on the X-ray structure of nhTMEM16 to position SCRD. However, to assess the reliability of the model it is important to know the identity percentage between the two sequences and whether or not sequence gaps were introduced for sequence alignment. It is stated in the text that 1-palmitoyl-2-stearoyl-sn-glycero-3-(phospho-L-serine) has been used for docking prediction, but apparently only the polar heads of PtdSer are represented in Figure 8. This should be stated in the text. Why are the acyl chains absent from the docking model? It is unclear from the model whether the SCRD interacts physically with PtdSer polar head.

Reviewer #3:

The paper sets up the problem in an artificial way, possibly reflecting the chronology of the authors' approach to the problem, but not relevant to a general readership: there is no question that members of the TMEM16 family are phospholipid scramblases based on the reconstitution and detailed testing of two fungal TMEM16s (afTMEM16 and nhTMEM16). The question being addressed is whether TMEM16F/ANO6 is a scramblase. This is important, as ANO6 is the ‘Scott syndrome’ protein whose functional deficiency prevents PS exposure on activated blood platelets resulting in a bleeding disorder. The ensuing studies of ANO6, and chimeric constructs comprising ANO6 sequences inserted into the non-scrambling ANO1, are very informative and this is what the authors should focus on.

The most important conclusion of the paper stems from the demonstration that a 15-amino acid stretch of ANO6, when transplanted into a comparable region of the ANO1 ion channel, confers phospholipid scramblase activity onto ANO1. Molecular modeling shows that this region falls within a membrane-facing hydrophilic groove defined by a recent crystal structure of a fungal TMEM16 homolog. This groove was suggested to be the lipid transport pathway, and the present data support this suggestion. The idea that ions ride along with phospholipids, or leak through the phospholipid pathway, is plausible but not completely firm. However, it seems to be the best that the authors can do with their experimental system.

Important specific issues:

1) The method of quantification of annexin binding is not explicitly described; as all conclusions about scrambling rest on this read-out, the method by which the graph shown in Figure 1 was generated needs to be described in detail.

2) The result of experiments without included caspase inhibitor should be shown in Figure 1. How was it verified that the caspase inhibitor worked?

3) Does the rate at which the Annexin signal is generated increase with the level of expression of ANO6 in the transient transfections? Or is the Annexin readout rate-limiting? This again demands details of quantification: what is the cell-to-cell variation in lag time and rate of signal increase in panel C?

4) The experiments in Figure 2 show that Ca^2+^, delivered by patch pipette, can promote currents and PS exposure (annexin binding). In panel C, how was F/F_o_ measured reliably as F_o_ would be ∼0 for the ANO1 case, and nowhere is it stated what time point was used to take F_o_ for the ANO6 case.

5) Figure 2: What is the explanation for the decrease in I/Imax for ANO1 vs. the increase in ANO6?

6) In the subsection headed “Is ANO1 a broken transporter?”: The *Saccharomyces* TMEM16 homolog has no detectable function; it does not transport ions or scramble lipids as per [55] and acts as a linker between the ER and PM (Manford et al. Dev Cell 2012). How does this fit the argument of ancient function of ANO proteins?

7) In the subsection headed “The ion conduction pathway”: The pathway for chloride in ANO1 requires amino acids located on the outside of the hydrophilic cleft, so how can it be started that this is an evolutionary remnant of the phospholipid pathway?

8) Figure 8: The SCRD is surprisingly asymmetric with respect to the membrane, lying mainly within the cytoplasmic leaflet. The authors offer no discussion about how this would offer a path for bidirectional movement of phospholipids.

9) The docking exercise was only performed with PS; however the fungal TMEM16 scramblases and the known activity of platelets and red blood cells move all phospholipids. What is the energetics of transport or PC? Or other major phospholipids?

---

## [Author Response]

*1) The authors should improve the annexin quantification and the correlation of its response with ANO6 expression*.

We have added extensive quantitative analysis of both Annexin-V binding and LactC2 binding as a measure of phospholipid scrambling (PLS). In response to this and other reviewers’ comments, Figures 1 and 2 have been completely reconfigured. Figure 1 now shows LactC2 binding. We quantify the percentage of cells that exhibit PLS, the expression of ANO1 and ANO6 by western blot, and the time course of LactC2 binding compared to AnnexinV binding. Figure 2 now quantifies AnnexinV binding relative to ANO6 expression and shows the time course of AnnexinV binding. Quantification has also been enhanced in Figure 6 (old Figure 5). Whereas previously we scored different chimeras as positive or negative for scrambling, we now provide the percentage of cells that scrambled. Figure 7 (old Figure 6) now includes plots of the percentage of cells that scramble and the time course of scrambling in one of the representative chimeras. New Figure 7—figure supplement 1 quantifies the ionic currents and PLS in two representative chimeras, one that scrambles and one that does not scramble.

2) The lag time between the ionic current and annexin binding can be explained by the non-linear response of annexin to PS density. A few experiments with Lactadherin, whose response to PS is more linear, could strengthen this part and validate the hypothesis that the ionic current is a consequence of the scramblase activity, not the other way round.

Figure 1 has been added to show scrambling experiments using lactadherin. We have quantified the percentage of cells scrambled and the time course, which is considerably slower than that of Annexin-V. Because the time course of LactC2 is slow, we have used Annexin for most of the experiments in the paper.

*3) In the present state, the lipid docking predictions are not detailed enough to be convincing. The authors should provide more information or delete all this part*.

The docking data have been deleted and Figure 8 (the homology model) has been modified to show additional views of the SCRD.

*4) In general, the paper would benefit from a major re-write to reduce jargon, and to state the problem and results clearly. Referencing would also benefit from a better appreciation of the lipid scrambling field*.

We have tried to edit the manuscript to make it more readable and to cite the scrambling literature more extensively. The paper has been essentially completely re-written to address the reviewers’ comments.

*There are also many other issues that you could address but the four above are the main ones that the reviewers consider essential*.

Reviewer #1:

Primary concerns:

*1) A trivial explanation for the difference in scramblase activity for GFP-ANO1 and GFP-ANO6 shown in*
Figure 1
*could be differences in the level of expression. Assuming the images were acquired with the same exposure settings, the authors could and should comment on the relative expression levels of their constructs.*

The expression levels of ANO6 were typically somewhat lower than ANO1 as judged both by immunofluorescence and by western blots. Western data has been added to Figure 1.

*2) For*
Figure 2*, it seems odd to me that the transfected ANO1 or ANO6 confer the major currents in the plasma membrane of the HEK cells. What kind of currents do untransfected cells display in these patch clamp experiments? Have these currents been measured and subtracted from the transfected cell data? Phosphatidylserine (PS) is also carrying charge across the membrane and the authors should indicate whether or not this contributes to the current they are measuring*.

We have previously published data showing that the currents in untransfected HEK cells under these conditions are negligible. For example, see online Figure 1 in [100] (PMID 22394518). The currents are small partly because the ionic conditions are chosen to eliminate K^+^ currents and HEK cells have no Na^+^ or Ca^2+^ currents. Under these conditions, background Cl^-^ currents are small.

*3) There is a tremendous amount of data squeezed into*
Figure 5
*and the impact/importance of the observations is somewhat obscured because of the complexity and density of information. In addition, the qualitative and more subjective nature of the “% scrambled cells*” *measure used in*
Figure 5
*and “yes/no*” *for scrambling activity in the table was unsatisfying because it is unclear how well the 1-6-1 chimeras have recapitulated the Ca*^*2+*^*-activated scramblase activity of ANO6. My primary recommendation is that the authors take their best 1-6-1 chimera carrying the SCRD and their best 6-1-6 chimera which loses SCR activity and perform the more quantitative kinetic assay as shown in*
Figure 2*. These data should be included in either*
Figure 5
*or*
Figure 6.

Figure 5 and Figure 6 (new Figures 6 and 7) have been reorganized for a more clear presentation and a considerable amount of new data have been added on quantification as indicated above. We have provided numbers of percentage of scrambled cells for the chimeras (instead of yes/no) and have expanded our characterization of the two best chimeras, as the reviewer suggests.

*4) The molecular docking of PS to the modeled structure in*
Figure 8
*is premature and should be removed. I don't think anyone has addressed the phospholipid specificity of ANO6 and I wouldn't expect a strong and specific binding site for PS in the membrane domain that mediates scrambling. My suggestion to the authors is to simply display the homology model and delete the description of their attempts to dock PS. Nothing meaningful was gained by this approach and the description seemed to detract from the presentation. One doesn't need this result to speculate on the translocation pathway in the Discussion section. I would prefer to see a zoomed in view of the SCRD in relationship to the proposed Ca*^*2+*^*-binding site for*
Figure 8.

We agree with the reviewer that the PS scramblase domain likely does not have a high affinity specific PS binding site. We have removed Figure 8 and replaced it with a view highlighting the SCRD and the location of the conserved Ca^2+^ binding site. We have also generated a zoomed in view of the SCRD and Ca^2+^ site to replace Figure 8.

Reviewer #2:

*The exposure of PtdSer on the external leaflet of the PM by phospholipid scrambling occurs in several biological processes and is an important cellular signaling mechanism. However, there is a great uncertainty about the identity of the phospholipid scramblases involved in such processes. This manuscript focuses on the lipid scrambling activity of ANO6/TMEM16F (hereafter ANO6), which has been proposed to be a Ca*^*2+*^*-activated scramblase. Yet, it has also been suggested that ANO6 is required for lipid scrambling by forming an ion channel but not by performing the scrambling activity itself. This manuscript convincingly shows that ANO6 is a ‘true’ scramblase and also indicates that the recorded ion channel activity of ANO6 is a consequence (a leak current) of phospholipid scrambling. In an elegant and very detailed manner the authors identify a minimal region in ANO6 required for PS scrambling. This is an excellent study*.

*1) Experiments include real-time monitoring of ion currents by patch clamp, and phospholipid scrambling using Annexin-V as indicator of exposed PtdSer. The authors find that ANO6 current is non-selective and allows passage of large ions such as NMDG+. The model concludes that ANO6 non-selective ion current coincides with phospholipid scrambling. Annexin-V has been widely used as a probe for exposed PtdSer, but Annexin-V response to PtdSer is not linear and requires a certain threshold of PtdSer to bind. Therefore, this probe is not well adapted for correlative real-time assays such as presented in*
Figure 2*. Notably, Annexin-V signal appears 3 min after the increase in current amplitude. This lag is commented in the text and the hypothesis raised by the authors is reasonable. However, is it possible to do the same experiment using a more robust PtdSer probe such as Lactadherin? Advantages of Lactadherin over Annexin-V have been listed in Shi J et al. 2006, a paper also cited by the authors*.

Lactadherin experiments have been added as Figure 1.

*2) The authors performed extensive sequence swapping studies to define the so-called scrambling domain (SCRD) in ANO6, which can confer the scrambling activity to ANO1. Mutations in SCRD prevented ANO6 from performing PtdSer scrambling. The manuscript presents a homology model of ANO6 based on the X-ray structure of nhTMEM16 to position SCRD. However, to assess the reliability of the model it is important to know the identity percentage between the two sequences and whether or not sequence gaps were introduced for sequence alignment. It is stated in the text that 1-palmitoyl-2-stearoyl-sn-glycero-3-(phospho-L-serine) has been used for docking prediction, but apparently only the polar heads of PtdSer are represented in*
Figure 8*. This should be stated in the text. Why are the acyl chains absent from the docking model? It is unclear from the model whether the SCRD interacts physically with PtdSer polar head*.

ANO6 (833 residues) was aligned to the TMEM16 sequence extracted from PDB ID 4wit (674 residues). These two sequences share 23% identity and 37.4% similarly (calculated by the BLSM62 algorithm). A total of 12 gaps were introduced, primarily at the N- and C-termini and in loops between secondary structural elements. Secondary structure prediction was incorporated into the alignment for Phyre2 (and all of the other modeling servers used for comparison including Swiss-Model and Tasser). There are no gaps in or near the SCRD, with nearest gaps located in extracellular loops between helices 3 and 4 (2 residue gap 24 residues away) and between helices 4 and 5 (1 residue gap 22 residues away). While overall alignment and homology model generation would benefit from a template with greater sequence identity, all modeling servers reported high confidence in the generated models with no variation in the placement of the SCRD between models.

The PtdSer docking has been deleted.

Reviewer #3:

*The paper sets up the problem in an artificial way, possibly reflecting the chronology of the authors' approach to the problem, but not relevant to a general readership: there is no question that members of the TMEM16 family are phospholipid scramblases based on the reconstitution and detailed testing of two fungal TMEM16s (afTMEM16 and nhTMEM16). The question being addressed is whether TMEM16F/ANO6 is a scramblase. This is important, as ANO6 is the ‘Scott syndrome’ protein whose functional deficiency prevents PS exposure on activated blood platelets resulting in a bleeding disorder. The ensuing studies of ANO6, and chimeric constructs comprising ANO6 sequences inserted into the non-scrambling ANO1, are very informative and this is what the authors should focus on*.

I am not sure I understand the full import of this comment. I interpret it to mean that the reviewer feels that the relationship of the ANO6 current is less important or interesting than the chimeric constructs. I tend to disagree, because there remains considerable uncertainty in the literature about the nature of the ANO6 current.

*The most important conclusion of the paper stems from the demonstration that a 15-amino acid stretch of ANO6, when transplanted into a comparable region of the ANO1 ion channel, confers phospholipid scramblase activity onto ANO1. Molecular modeling shows that this region falls within a membrane-facing hydrophilic groove defined by a recent crystal structure of a fungal TMEM16 homolog. This groove was suggested to be the lipid transport pathway, and the present data support this suggestion. The idea that ions ride along with phospholipids, or leak through the phospholipid pathway, is plausible but not completely firm. However, it seems to be the best that the authors can do with their experimental system*.

I agree fully with the reviewer’s comments. In some ways, this idea is not completely compelling, but yet it does fit the data that we have.

*Important specific issues*:

*1) The method of quantification of annexin binding is not explicitly described; as all conclusions about scrambling rest on this read-out, the method by which the graph shown in*
Figure 1
*was generated needs to be described in detail.*

We have expanded our quantification extensively and have described our methods in detail.

*2) The result of experiments without included caspase inhibitor should be shown in*
Figure 1*. How was it verified that the caspase inhibitor worked?*

Figures 1 and 2 have been replaced with new data and the caspase inhibitor data have been removed. We feel that the caspase inhibitor is a side-issue and that it is not necessary for this paper to address the question whether caspase is required for ANO6 scrambling.

3) Does the rate at which the Annexin signal is generated increase with the level of expression of ANO6 in the transient transfections? Or is the Annexin readout rate-limiting? This again demands details of quantification: what is the cell-to-cell variation in lag time and rate of signal increase in panel C?

We have expanded our quantification extensively as noted above.

*4) The experiments in*
Figure 2
*show that Ca2*^*+*^*, delivered by patch pipette, can promote currents and PS exposure (annexin binding). In panel C, how was F/ F*_*o*_
*measured reliably as F*_*o*_
*would be ∼0 for the ANO1 case, and nowhere is it stated what time point was used to take F*_*o*_
*for the ANO6 case*.

F/F_o_ fluctuated around 1 (not zero) for ANO1. F_o_ for ANO6 was taken immediately upon establishing whole-cell recording. This has been added to the figure legend.

*5)*
Figure 2: *What is the explanation for the decrease in I/Imax for ANO1 vs. the increase in ANO6?*

We have previously published details of the Ca^2+^-dependent rundown of the ANO1 current. The reference to these data of [101] is cited in the text. The mechanism of this remains unknown.

*6) In the subsection headed “Is ANO1 a broken transporter?”: The* Saccharomyces *TMEM16 homolog has no detectable function; it does not transport ions or scramble lipids as per*
[55]
*and acts as a linker between the ER and PM (Manford et al. Dev Cell 2012). How does this fit the argument of ancient function of ANO proteins?*

This is a great question. My intuition tells me that the “linker” function of Ist2p is somehow related to transport of lipids from the ER to the plasma membrane. We have added the following: “The *Saccharomyces* ANO homolog, Ist2p, has been shown to play a role […]. ER-plasma membrane junctions are known to be involved in non-vesicular sterol transport from ER to plasma membrane.”

*7) In the subsection headed “The ion conduction pathway”: The pathway for chloride in ANO1 requires amino acids located on the outside of the hydrophilic cleft*, *so how can it be started that this is an evolutionary remnant of the phospholipid pathway?*

I am not sure I understand this question. The amino acids we identify are predicted to be at the extracellular surface of the protein and form the extracellular entryway into the conduction pore, which we think may coincide in part with the hydrophilic cleft. Obviously, there are many questions we cannot answer with the data available.

*8)*
Figure 8*: The SCRD is surprisingly asymmetric with respect to the membrane, lying mainly within the cytoplasmic leaflet. The authors offer no discussion about how this would offer a path for bidirectional movement of phospholipids*.

Certainly, there is the likelihood that other parts of the protein or accessory proteins may play additional roles, but at this point in time we have little insight into this question.

9) The docking exercise was only performed with PS; however the fungal TMEM16 scramblases and the known activity of platelets and red blood cells move all phospholipids. What is the energetics of transport or PC? Or other major phospholipids?

This section has been deleted.